# SGK1 inhibition in glia ameliorates pathologies and symptoms in Parkinson disease animal models

Oh-Chan Kwon[1,2,3,†] , Jae-Jin Song[1,2,†], Yunseon Yang[1,2,3,†], Seong-Hoon Kim[1,2,3], Ji Young Kim[1,2,3], Min-Jong Seok[1,2,3], Inhwa Hwang[4], Je-Wook Yu[4] , Jenisha Karmacharya[5], Han-Joo Maeng[5], Jiyoung Kim[6], Eek-hoon Jho[6] , Seung Yeon Ko[1,2,3], Hyeon Son[1,2,3], Mi-Yoon Chang[1,2,*] & Sang-Hun Lee[1,2,3,**]

## Abstract

Astrocytes and microglia are brain-resident glia that can establish harmful inflammatory environments in disease contexts and thereby contribute to the progression of neuronal loss in neurodegenerative disorders. Correcting the diseased properties of glia is therefore an appealing strategy for treating brain diseases. Previous studies have shown that serum/ glucocorticoid related kinase 1 (SGK1) is upregulated in the brains of patients with various neurodegenerative disorders, suggesting its involvement in the pathogenesis of those diseases. In this study, we show that inhibiting glial SGK1 corrects the pro-inflammatory properties of glia by suppressing the intracellular NFκB-, NLRP3-inflammasome-, and CGAS-STING-mediated inflammatory pathways. Furthermore, SGK1 inhibition potentiated glial activity to scavenge glutamate toxicity and prevented glial cell senescence and mitochondrial damage, which have recently been reported as critical pathologic features of and therapeutic targets in Parkinson disease (PD) and Alzheimer disease (AD). Along with those anti-inflammatory/neurotrophic functions, silencing and pharmacological inhibition of SGK1 protected midbrain dopamine neurons from degeneration and cured pathologic synuclein alpha (SNCA) aggregation and PD-associated behavioral deficits in multiple *in vitro* and *in vivo* PD models. Collectively, these findings suggest that SGK1 inhibition could be a useful strategy for treating PD and other neurodegenerative disorders that share the common pathology of glia-mediated neuroinflammation.

**Keywords** glia; neuroinflammation; Parkinson's disease; serum/glucocorticoid related kinase 1; synuclein alpha
**Subject Category** Neuroscience

## Introduction

Parkinson disease (PD) is a common neurodegenerative disorder characterized by the degeneration of dopamine (DA) neurons in the substantia nigra (SN) of the midbrain and toxic intra-neuronal inclusion of misfolded synuclein alpha (Lewy bodies and neurites). Studies have shown that PD's pathologic features are caused by mitochondrial damage/dysfunction, endoplasmic reticulum stress, and defects in vesicular trafficking, synaptic recycling, and autolysosomal pathways in midbrain DA (mDA) neurons (Nguyen *et al*, 2019). Along with those intra-neuronal dysfunctions, persistent inflammation mediated primarily by neighboring glia commonly underlies PD and other neurodegenerative disorders (Yan *et al*, 2014; Guzman-Martinez *et al*, 2019).

Astrocytes and microglia are brain-resident glial cells that typically play a homeostatic and neurotrophic role supporting neuronal cell survival and functioning and maintaining brain homeostasis in other ways (Horner & Palmer, 2003; Nedergaard *et al*, 2003; Molofsky *et al*, 2012). However, the naïve homeostatic properties of glia are compromised in disease contexts, and they instead establish harmful inflammatory brain environments by secreting inflammatory mediators (frequently annotated as M1 [microglia] and A1 [astrocyte] activation; Neumann *et al*, 2009; Liddelow *et al*, 2017a). The idea of polarizing pathologic M1(A1)-type glia back into their neurotrophic/homeostatic forms (M2 and A2 types) to treat neurodegenerative diseases for which there are no disease-modifying therapies has gained momentum (Hamby & Sofroniew, 2010).

A previous study demonstrated that nuclear receptor subfamily 4 group A member 2 (NR4A2, also known as NURR1), originally known as a transcription factor specific for developing and adult mDA neurons, can also be expressed in astrocytes/microglia in response to toxic insults and that Nurr1-expressing glia protect neighboring mDA neurons by prohibiting the synthesis and release

1   Department of Biochemistry and Molecular Biology, College of Medicine, Hanyang University, Seoul, Korea
2   Hanyang Biomedical Research Institute, Hanyang University, Seoul, Korea
3   Graduate School of Biomedical Science and Engineering, Hanyang University, Seoul
4   Korea Department of Microbiology and Immunology, Institute for Immunology and Immunological Diseases, Brain Korea 21 PLUS Project for Medical Science, Yonsei University College of Medicine, Seoul, South Korea
5   College of Pharmacy, Gachon University, Incheon, Korea
6   Department of Life Science, University of Seoul, Seoul, Korea
    *Corresponding author. Tel: +82 2 2220 0620; E-mail: mychang@hanyang.ac.kr
    **Corresponding author. Tel: +82 2 2220 0625; Fax: +82 2 2220 2422; E-mail: leesh@hanyang.ac.kr
    †These authors contributed equally to this work

of pro-inflammatory cytokines from glial cells (Saijo *et al*, 2009). This Nurr1-mediated anti-inflammatory action in glia has been manifested in various brain disease contexts (Loppi *et al*, 2018; Popichak *et al*, 2018; Montarolo *et al*, 2019; Shao *et al*, 2019). Other studies have shown that forkhead box A2 (FOXA2; also known as HNF3B) is a potent cofactor that synergizes the anti-inflammatory role of Nurr1 in glia (Oh *et al*, 2015; Song *et al*, 2018). Studying intracellular anti-inflammatory pathways downstream of Nurr1 + Foxa2 in glia could be valuable for identifying reliable therapeutic targets to treat neurodegenerative disorders, including PD.

Serum/ glucocorticoid related kinase 1 (SGK1) is a gene first identified as upregulated by serum and glucocorticoids in rat mammary tumor cells (Firestone *et al*, 2003). SGK1 expression is widely detected in the brain, and it is increased in pathologic conditions such as Rett syndrome (Nuber *et al*, 2005), Alzheimer disease (Chun *et al*, 2004; Lang *et al*, 2010; Zhang *et al*, 2018), multiple sclerosis (Wang *et al*, 2017), amyotrophic lateral sclerosis (Schoenebeck *et al*, 2005), and neuropathic pain (Geranton *et al*, 2007; Peng *et al*, 2013). The level of SGK1 is relatively higher in the midbrain than in other brain regions, and upregulation of SGK1 coincides with the onset of DA neuron death in a model of PD (Iwata *et al*, 2004; Stichel *et al*, 2005), collectively suggesting that SGK1 plays pathogenic roles in PD and other neurodegenerative disorders. However, SGK1 role in CNS is not clearly identified. In this study, we show that Sgk1 is a molecule that is negatively regulated by Nurr1 (N) and Foxa2 (F) in glial cells. We also show that glial inhibition of SGK1 mediates the anti-inflammatory functions of N + F. In addition to its anti-inflammatory effects, SGK1 inhibition in glia potentiates the neuroprotective functions of glia and prevents glial cell senescence, which has recently been reported as a critical pathologic feature of and therapeutic target in AD and PD (Bussian *et al*, 2018; Chinta *et al*, 2018). Using multiple *in vitro* and *in vivo* PD models, we show that silencing and pharmacological inhibition of SGK1 are effective therapeutic tools for treating PD.

# Results

## SGK1 inhibition in glia suppresses NFκB-mediated pro-inflammatory cytokine gene expression

We and other research groups have previously shown that the midbrain factors N and F exert anti-inflammatory effects in glia by inhibiting the transcription of pro-inflammatory cytokines (Saijo *et al*, 2009; Oh *et al*, 2015; Song *et al*, 2018). In our microarray data (accession no. GSE145489), one of the genes that N and F most synergistically downregulated in glia cultured from mouse cortices was *Sgk1* (reduced 9.3-fold by N and F in combination; Fig 1A and B). We further observed a similar reduction in SGK1 expression in N + F-expressing astrocytes derived from a mouse ventral midbrain (VM) in our RNA-sequencing data (RNA-seq; accession no. GSE106216; Fig 1C), which we validated in VM-derived glial cultures (GFAP$^+$ astrocytes > 70%, IBA1$^+$ microglia < 15%, exposure to $H_2O_2$, 250 μM, 4 h) using qPCR analyses (Fig 1D). SGK1 inhibition attracted our interest because SGK1 could regulate NFκB signal, the central intracellular pathway responsible for inflammatory cytokine transcription (Lawrence, 2009; Liu *et al*, 2017). Specifically, SGK1 is suggested to phosphorylate IκB kinase subunit beta

(IKKB), which activates NFκB signaling by releasing NFκB from the inhibitory IκB-NFκB complex(Lang & Voelkl, 2013). Indeed, Western blot (WB) analyses showed that downregulating SGK1 proteins by forcing the expression of N and F in cultured VM-derived glial cells was followed by a decrease in phosphorylated (activated) IKKB and IκB (Fig 1E). Consequently, compared to the mock-transduced control glia, the higher proportion of the NFκB in N- and F-transduced glia was inactively trapped in the complex associated with IκB in cytosol (Fig 1F). The inactivation of NFκB signaling was further manifested by decreased levels of phosphorylated (activated) p65 (RELA: RELA proto-oncogene, NF-kB subunit), a major NFκB component, and decreased levels of RELA (p65) proteins trans-localized into the nucleus (Fig 1E). The N + F-mediated inactivation of NFκB signals in glia was recovered by the forced expression of SGK1 (Fig 1G), and the expression of pro-inflammatory cytokines reduced by N + F treatment returned to levels similar to those in the control glia (Fig 1H). NFκB activation induced by lipopolysaccharide (LPS, 1 mM, 24 h), a toll-like receptor 4 ligand, greatly subsided in cultured glial cells in the presence of a SGK1 inhibitor (GSK-650394) (Fig 1I). As expected, SGK1 inhibition using treatment with siRNA, sh-RNA, and specific inhibitors (EMD-638683, GSK-650394) consistently downregulated the key pro-inflammatory cytokines IL-1β, TNFα, IL-6, and iNOS (Fig 1J–M), whereas the overexpression of SGK1 had the opposite effect (Fig 1N). These findings collectively indicate that N + F-mediated anti-inflammatory functions in glia are substantially attained by blocking SGK1 signaling (Fig 1O), and thus, SGK1 inhibition could be used to treat neurodegenerative disorders in which neuroinflammation is a common underlying pathology.

## SGK1 inhibition-mediated transcriptome changes are associated with immune/inflammation reaction and glial activation/polarization

The main purpose of this study was to test whether SGK1 inhibition in glial cells could be used to develop a therapy for PD. Therefore, glial cells were cultured from rodent VMs, which is primarily affected in PD. VM-glial cultures of GFAP$^+$ astrocytes (60–70% of total cells) and Iba1$^+$ microglia (5–15%, cf > 19.7 ± 1.7% counted in mouse adult midbrain) were used throughout this study unless otherwise noted because astroglia and microglia interact closely to prevent or exacerbate disease pathogenesis (Jha *et al*, 2019), and thus, the consequences of their interactions in mixed glia, not the isolated individual actions of astrocytes and microglia, are the target in developing therapeutic interventions for brain disease. To gain further insight into how SGK1 inhibition affects glial functions, we performed an RNA-seq analysis (GSE145490) in primary VM-derived glial cultures with and without the SGK1 inhibitor GSK-650394. Consistent with the SGK1 inhibition effects shown in Fig 1, genes downregulated by the SGK1 inhibitor (FPKM > 1, > 2.6 fold change [FC]) were found in the gene ontologies associated with "inflammatory reaction" and the "NFκB pathway" (highlighted in red in Fig 2A and B). Actually, 7 of the top 10 ontologies of the downregulated genes were related to inflammatory/immune reactions. The enrichment of differentially regulated genes (DEGs) in inflammatory/immune pathways was further confirmed by gene set enrichment analyses (Fig 2C). Furthermore, immune/inflammatory genes were the most significantly downregulated by the inhibitor treatment

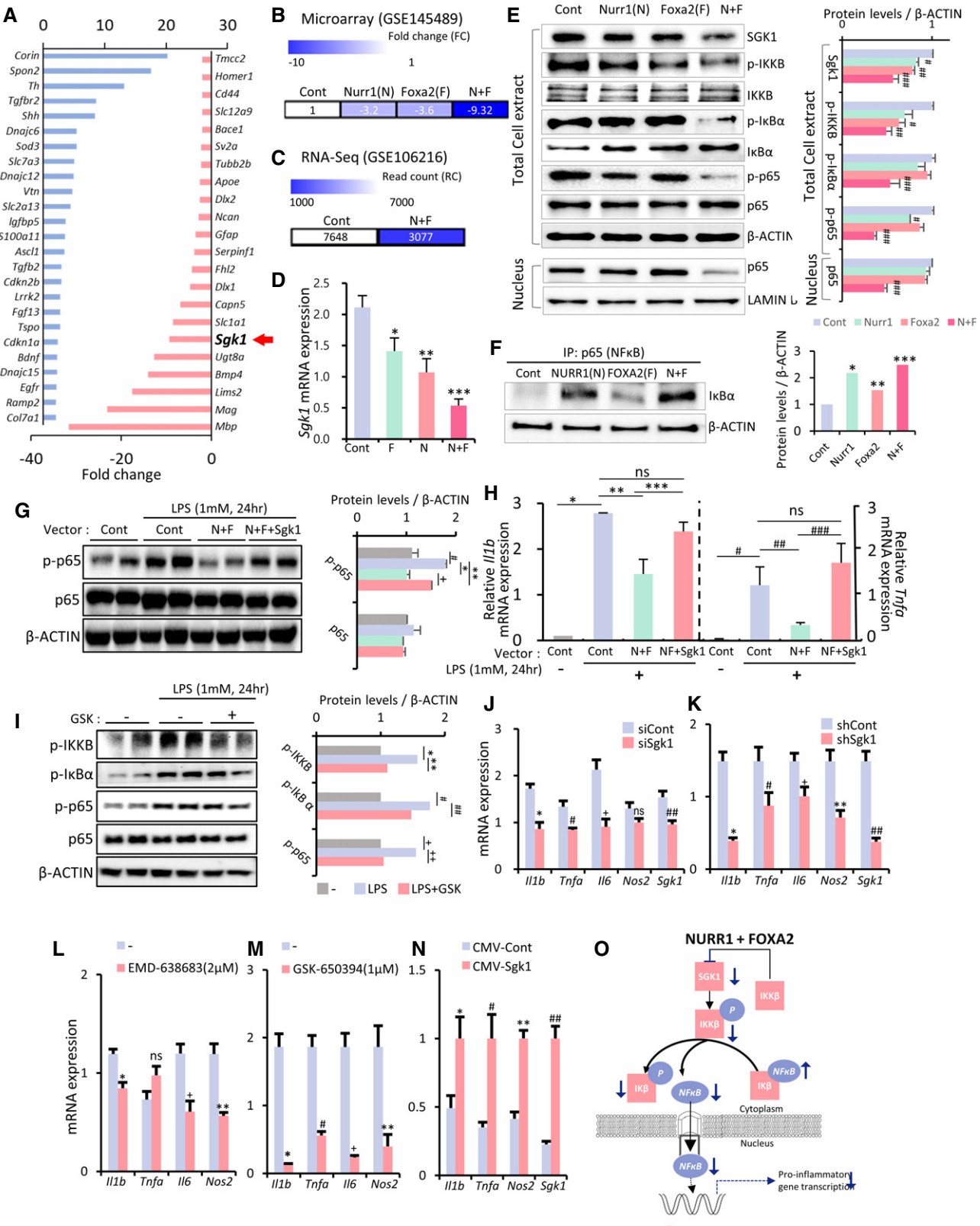

**Figure 1.**

**Figure 1.  SGK1 inhibition is the key mediator for Nurr1 + Foxa2-induced anti-inflammatory function in glial cells.**

A–D    Nurr1 (N) and Foxa2 (F) synergistically downregulate SGK1 expression in cultured glia. (A) Up- and downregulated genes in the microarray data for the cultured
glia transduced with N + F (vs. mock-transduced control). *Sgk1* is marked with an arrow in the top downregulated genes. (B–D) Effect of N and F on SGK1
expression in glial cells shown in the microarray (B), RNA-seq (C), and RT–PCR (D) analyses. Data in (B and C) represent values of fold change (FC) or read count
(RC) with color intensities. *n* = 3. One-way ANOVA. Data are represented as mean ± SEM. Significantly different at *P* = 0.0493*, 0.0092**, 0.0009***.

E–I    Downregulation of SGK1 is responsible for N + F function to inhibit NFκB-mediated pro-inflammatory cytokine expression. (E) WB analysis exhibiting N + F-
induced inhibition of NFκB intracellular signaling in cultured VM-glia. The N + F-mediated downregulation of NFκB signaling is attained by inhibiting IκB
phosphorylation (E) and thus blocking the release of NFκB from the NFκB-IκB inhibitory complex (F). Data are represented as mean ± SEM. *n* = 3 independent
experiments; one-way ANOVA with Tukey-analysis. *P* = 0.015#, 0.002##, 1.33E-12### (sgk1), 0.036#, 0.00004## (p-IKKB). 0.00007### (p-IkBα). 0.005#, 2.11E-09###
(p-p65). 1.93E-06### (p65 level in nucleus) in graph E. *P* = 6.44E-09*, 0.00004**, 4.45E-10*** in graph F. (G and H) N + F-mediated downregulation of NFκB
signaling (phosphorylation of p65, p-p65, G) and pro-inflammatory cytokine expression (IL-1β, TNFα, H) were alleviated by forced SGK1 expression in the VM-glial
cultures. Data are represented as mean ± SEM. *n* = 3 independent cultures; two-way ANOVA with Bonferroni *post hoc* analysis. Significantly different at
*P* = 0.014#, 0.009*, 0.009**, 0.003+ (p-p65) in graph G and *P* = 1.79E-06*, 0.0003**, 0.004***, 0.0068#, 0.0392##, 0.0027### in graph H. (I) SGK1 inhibition blocks
glial NFκB signaling. Data are represented as mean ± SEM. *n* = 4–6. One-way ANOVA. Significantly different at *P* = 0.002*, 0.00004**, 0.00001, 3.01E-10##,
3.78E-06+, 1.59E-07++ in graph I.

J–N    Glial pro-inflammatory cytokine expression regulated by SGK1 inhibition (J–M) and overexpression (N). Data are represented as mean ± SEM. *n* = 3 independent
cultures; unpaired Student's *t*-test. Significantly different at *P* = 0.0041*, 0.0010#, 0.0010+, 0.0067## in graph J and *P* = 0.0002*, 0.0345#, 0.0023+, 0.0012**,
0.0001## in graph K and *P* = 0.0367*, 0.0042+, 0.0003** in graph L and *P* = 0.0001*, 0.0009#, 0.0002+, 0.0082** in graph M and *P* = 0.0367*, 0.0059#, 0.0005**,
0.0006## in graph N.

O    Schematic summary to show how SGK1 inhibition mediates the N + F-induced anti-inflammatory action in glia.

(Fig 2D and E). Using a 2-FC cutoff and 1% false discovery rate (FDR), 726 genes downregulated by the SGK1 inhibitor were also downregulated by N + F, and the overlapping genes were also found in the gene categories of immune/inflammation and the NFκB pathway (Appendix Fig S1A, C, E), further confirming that N + F inhibition of glial inflammation via the NFκB pathway is mediated by inhibiting SGK1 intracellular signals in glial cells.

In the RNA-seq data, the expression of all the M1-specific markers reported (Ka *et al*, 2014) was downregulated by treatment with the SGK1 inhibitor. The expression of the M2 markers *Il10, Ccl22, Egf, Fizz1, and Il2ra* was upregulated, while five of them were downregulated (Fig 2F), indicating SGK1 inhibition establishes its own microglial transcriptome signature mainly by inhibiting M1 marker transcription. Out of 13 pan-reactive astrocyte markers (Liddelow *et al*, 2017b), 10 were upregulated in the glial cultures treated with the inhibitor. However, there was no clear A1–A2 trend in the astrocyte-specific marker expression patterns, indicating that SGK1 inhibition induces a distinct glial reactivity profile, as also shown in other recent studies (Marschallinger *et al*, 2020; Smith *et al*, 2020; Zhou *et al*, 2020).

## NLRP3-inflammasome- and CGAS-STING-mediated inflammatory pathways in glia are suppressed by SGK1 inhibition

Downregulated inflammatory genes in the RNA-seq data included the key components of the NLR family pyrin domain containing 3 (NLRP3) inflammasome signal (Fig 3A), which is required for the activation of IL1B and IL18 and thus regarded as the archetypical molecular driver of inflammatory response (reviewed in ref. Afonina *et al*, 2017). The decreased mRNA expression of *Nlrp3*, PYD and CARD domain containing (*Pycard*, also known as *Asc*) apoptosis-associated speck-like protein containing a caspase recruitment domain (ASC), and procaspase-1 was validated in independent cultures treated with the SGK1 inhibitor, sh-Sgk1, and si-Sgk1 (Fig 3B), indicating that SGK1 signals are involved in NRLP3 inflammasome activation in glial cells. Note that the transcriptional control of NLRP3 inflammasome components is considered rate limiting in the activation of the inflammasome (Huai *et al*, 2014; Ising *et al*, 2019). To determine whether SGK1 inhibition could indeed prevent the activation of the

glial NRLP3 inflammasome pathway, VM-derived glial cells cultured in the presence or absence of the SGK1 inhibitor (for 4 days) were primed with LPS (0.25 µg/ml, 3 h), and then followed by ATP treatment (2.5 mM, 30 min). In the control cultures, the sequential LPS-ATP treatment robustly induced the activation of caspase1 and the secretion of IL1B, as determined by the detection of active caspase1 (p10) and mature/activated IL1B protein levels in the control cell culture supernatants (Fig 3C). By contrast, activated and secreted caspase1 and IL1B were markedly decreased to almost undetectable levels in the cultures treated with the SGK1 inhibitor. The reduced activation of the NLRP3 inflammasome through SGK1 inhibition was further confirmed by the ELISA detection of IL1B in the culture medium (Fig 3D). The activation of the NRLP3-inflammasome has lately been reported as a central pathogenic contributor to neurodegenerative diseases (Voet *et al*, 2019), including AD (Heneka *et al*, 2013; Venegas *et al*, 2017; Ising *et al*, 2019) and PD (Mao *et al*, 2017; Lee *et al*, 2018), further indicating that SGK1 inhibition in glia could be a therapeutic intervention for those disorders.

We also observed that the inflammatory genes downregulated in the glial cultures treated with the SGK1 inhibitor included components of the cyclic GMP-AMP synthase (CGAS)-stimulator of interferon response cGAMP interactor 1 genes (STING) pathway (Fig 3E and F), which has recently been identified as another critical pathogenic contributor to PD caused by Parkin and PINK1 mutations (Sliter *et al*, 2018). To assess the regulation of CGAS-STING signaling by glial SGK1 inhibition, VM-glia were treated with $H_2O_2$ (250 µM) and LPS (1 mM), chemicals reported to activate that pathway (Gehrke *et al*, 2013; Wang *et al*, 2019). In this treatment condition, the activation of the STING pathway was evidenced by increased levels of CGAS, STING, phospho-TBK1, and phospho-IRF3 proteins (Fig 3G). Treatment with the SGK1 inhibitor significantly blunted the increased activation of those CGAS-STING signal molecules in the cultured VM-glia (Fig 3G) and drastically decreased the expression of *Ifnb1* mRNA, a final product of the CGAS-STING pathway (Fig 3H). Collectively, these findings show that SGK1 inhibition in glia strongly potentiates anti-inflammatory functions by suppressing the NFκB, NLRP3-inflammasome, and CGAS-STING signal pathways.

### SGK1 inhibition in glia upregulates glutamate clearance capacity

NFκB signaling activated in astrocytes has been shown to suppress the expression of the glutamate transporter SLC1A2 (solute carrier family 1 member 2; also known as EAAT2, GLT-1; Fine *et al*, 1996; Su *et al*, 2003; Boycott *et al*, 2008; Jiang *et al*, 2019), the protein that reduces glutamate toxicity by clearing extracellular glutamate and thus protect neuronal cells from excitatory toxicity (Hansson *et al*, 2000). Consistently, along with the downregulated NFκB signaling in the glia treated with the SGK1 inhibitor (Fig 1I), a > 3.7-fold

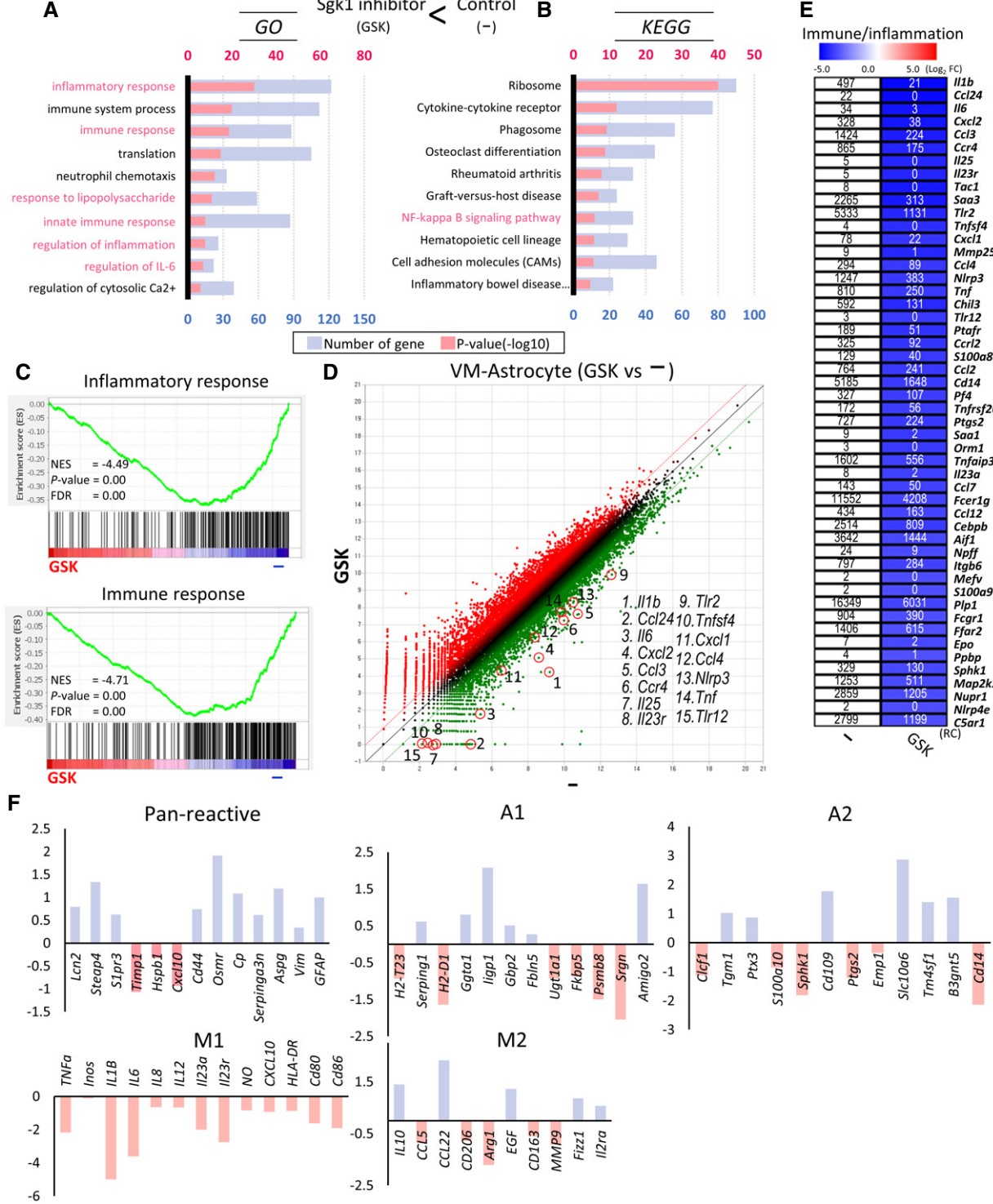

**Figure 2.**

**Figure 2. SGK1 inhibition-mediated transcriptome changes associated with immune/ inflammation reaction and microglial polarization.**

In the RNA-seq data (GSE145490, normalized read count (RC) > 1, > 2.6-fold, 3,000 genes), genes downregulated in the VM-glia treated with the SGK1 inhibitor GSK-650394 vs. the vehicle (DMSO)-treated control were analyzed.

A, B   Top gene ontologies (GOs) and KEGG pathways of the genes downregulated in VM-glia by the SGK1 inhibitor treatment. The purple bar indicates the number of genes under the designated GO term/KEGG pathway. The red bar indicates the p-value, and the negative log of the p-value (bottom) is plotted on the X-axis.
C   Gene set enrichment analysis (GESA) for inflammatory and immune responses.
D   Scatterplots for the differentially expressed genes (DEGs) highlighting the inflammatory/immune genes that are most significantly downregulated in the inhibitor-treated cultures.
E   Heatmap for selected inflammatory genes in the SGK1 inhibitor-treated and inhibitor-untreated glial cultures. Data represent the RC values (inside box) and $\log_2$ [SGK1 inhibitor-treated/control] (color intensity).
F   Microglia and astrocyte reactivation-specific marker expressions in the RNA-seq data.

expression increase of the glutamate transporters *Slc1a3* (solute carrier family 1 member 3; also known as, GLAST, EAAT1) and *Slc1a2* was seen in the RNA-seq data (GSE145490) by the SGK1 inhibitor treatment (Appendix Fig S2A). The glutamate transporter expression was further validated in independent cultures treated with the SGK1 inhibitors (Appendix Fig S2B). In addition, we found that SGK1 inhibitor treatment significantly increased glutamate uptake activity in cultured VM-glia (Appendix Fig S2C). It is noted that astrocytic GLT-1 (SLC1A2) deficiency caused mDA neuron death and PD symptoms (Zhang *et al*, 2020), while increased GLT-1 expression has also been shown to have beneficial and therapeutic effects (Kobayashi *et al*, 2018). Thus, these findings collectively indicate that SGK1 inhibition enhances therapeutic capacity of glia via upregulation of glutamate clearance as well as the anti-inflammatory activities.

## Mitochondrial damage in glia is blunted by treatment with the SGK1 inhibitor

Mitochondrial defects and dysfunction are the leading cause of neurodegeneration (reviewed in ref. Johri & Beal, 2012; Lezi & Swerdlow, 2012). Neurons are the central cell type to be studied for the link between mitochondrial pathology and neurodegeneration in PD and other disorders. The importance of mitochondrial biology in glia has recently come into the spotlight (reviewed in ref. McAvoy & Kawamata, 2019) especially because of reports demonstrating the role of healthy mitochondria from astrocytes in rescuing neurons from injury (Hayakawa *et al*, 2016) and demonstrating the pathologic contribution of damaged and fragmented mitochondria in glia to M1(A1) polarization and thus the propagation of inflammatory neurodegeneration (Joshi *et al*, 2019). The NLRP3-inflammasome and CGAS-STING pathways, which were blocked by SGK1 inhibition in glial cells (Fig 3), are closely linked to mitochondrial damage and dysfunction (Sorbara & Girardin, 2011; Heid *et al*, 2013; Liu *et al*, 2016; Maekawa *et al*, 2019). In addition, the upregulated expression of PGC1α, the master regulator for mitochondrial biogenesis (Wu *et al*, 1999; Finck & Kelly, 2006), was detected in the RNA-seq data from glia treated with the SGK1 inhibitor (Fig 4A) and confirmed using qPCR (Fig 4B) and WB analyses (Fig 4C) in cultures treated with GSK-650394 or EMD-638683, indicating that SGK1 inhibition could affect mitochondrial functioning/metabolism in glia. As anticipated, the MitoTimer assay (Ferree *et al*, 2013; Hernandez *et al*, 2013) showed that the ratio of healthy or newly synthesized mitochondria (green fluorescence) to damaged or aged mitochondria (red) was greater in glia treated with the SGK1 inhibitor than in control glia (Fig 4D). Treating cultured VM-glia with H2O2 + CCCP (mitochondrial toxin) increased the reactive oxygen species (ROS)

in the mitochondria (MitoSox) (Fig 4E) and reduced the mitochondrial membrane potential (JC-1) (Fig 4F). The toxin-induced mitochondrial damage in glia was significantly reduced in the presence of the SGK1 inhibitor GSK-650394 (Fig 4E and F). Along with the increased number of healthy mitochondria and reduced mitochondrial stress, intracellular levels of ATP, an indicator of mitochondrial function (oxidative phosphorylation), increased in the glia treated with GSK-650394 in a dose-dependent manner (Fig 4G). These findings suggest that SGK1 inhibition improves glial mitochondrial health and function, which could contribute to the neuroprotective functions of glia (Hayakawa *et al*, 2016) and prevent glia-mediated propagation of disease pathology (Joshi *et al*, 2019).

## Effects of SGK1 inhibition on glial cell senescence

Recent studies have shown that cellular senescence in glia is another key pathologic feature of neurodegenerative disorders, and thus, clearing or preventing senescent glial cells has been suggested as a therapeutic avenue for PD (Chinta *et al*, 2018) and AD (Bhat *et al*, 2012; Bussian *et al*, 2018; Zhang *et al*, 2019). In the same context, it has been acknowledged that astrocytic oxidative stress, a leading cause of cell senescence, contributes to PD pathogenesis (reviewed in ref. Rizor *et al*, 2019). The following experimental evidence from our and other studies strongly indicates that SGK1 inhibition could rescue glial cells from senescence and oxidative stress. First, inflammation, shown in this study to be reduced by SGK1 inhibition, is closely linked to increased ROS generation: specifically, the pro-inflammatory cytokine TNFα binds to TNFR and then produces intracellular ROS via NADPH oxidase 1/2 (reviewed in ref. Blaser *et al*, 2016). Second, our RNA-seq data show that treating glia with an SGK1 inhibitor downregulated the prooxidant *Cyba*, *Ncf2*, *Recql4*, *Bax,* and *Fdxr* and senescence-associated secretory phenotype (SASP) genes (Fig 4H). Third, SGK1 inhibitor treatment repressed the formation of mitochondrial ROS (MitoSox), a major source of intracellular ROS, in cultured glia (Fig 4E). Therefore, we assessed whether oxidative stress and cellular senescence in glial cells could be alleviated by SGK1 inhibition. As expected, oxidative stress, estimated using dihydrodichlorofluorescein diacetate (DCF), a dye detecting cellular ROS, was greatly reduced in H2O2-treated glial cultures by treatment with the SGK1 inhibitor (Fig 4I). We further observed that senescence-associated β-galactosidase (SA-β-gal) activity, the hallmark senescence biomarker, was promoted in glia exposed to H2O2 and significantly rescued in the presence of the SGK1 inhibitor (Fig 4J). In addition, in a paraquat-induced glial senescence model (Chinta *et al*, 2018), a WB analysis showed that the herbicide-induced LMNB1 (Lamin B1)

loss, a senescence-associated biomarker (Freund *et al*, 2012), was alleviated in glial cultures treated with the SGK1 inhibitor, and levels of the pro-senescence proteins IL6, MMP3, and p16 (CDKN2A) also decreased (Fig 4K). Reduced secretion of IL6, a cytokine of SASP, from cultured glia treated with the SGK1 inhibitor was further demonstrated using ELISA (Fig 4L). Another key feature of senescent cells is their inherent sensitivity to senolytic drugs. When

cultured glia were exposed to two different senolytic drugs that target disparate molecular pathways, they demonstrated dose-dependent loss of cell viability (Fig 4M). SGK1 inhibitor treatment endowed the glia with significant resistance to the senolytic drugs. These findings collectively suggest that SGK1 inhibition in PD brains could reduce disease pathology by ameliorating glial oxidative stress and senescence.

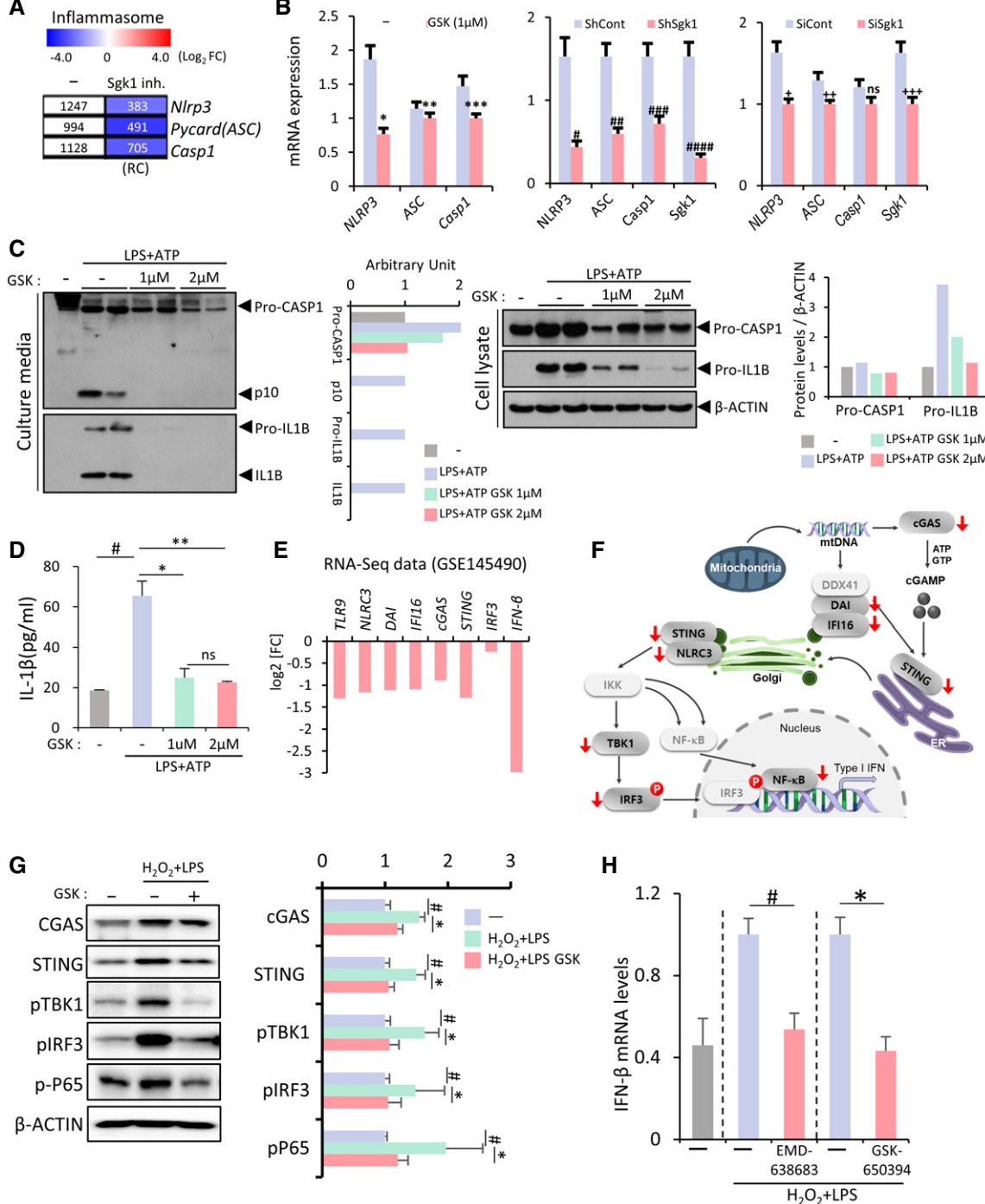

**Figure 3.**

Figure 3. Glial NLRP3-inflammasome and CGAS-STING inflammatory pathways downregulated by SGK1 inhibition in VM-glial cultures.

A, B  mRNA expression of the NLRP3-inflammasome components *Nlrp3*, *Pycard (Asc)*, and *Caspase1*. Decreased expression of the components shown in the RNA-seq (A) was confirmed using qPCR analyses in other independent VM-glial cultures treated with the inhibitor (GSK-650394) and sh-Sgk1 and si-Sgk1 (vs. DMSO, sh-control, and si-control, respectively) (B). n = 3; Student's *t*-test. Data are expressed as mean ± SEM. Significantly different at P = 0.0018*, 0.0405**, 0.0039***, 0.0002[#], 0.0033[##], 0.0099[###], 0.0001[####], 0.0056[+], 0.0398[++], 0.0067[+++] in graph B.

C  Immunoblot analysis to assess NLRP3 inflammasome activation. To activate the inflammasome pathway, VM-derived glial cells were treated with LPS (0.25 μg/ml, 3 h) and then ATP (2.5 mM, 30 min) in the presence or absence of the SGK1 inhibitor GSK-650394. Two days later, intracellular and released levels of pro- and activated caspase-1 and IL-1β proteins were determined in the culture media and cell lysates, respectively.

D  Released level of IL-1β (activated) was further quantified using ELISA. n = 3; ANOVA. Significantly different at p = 0.0002[#], 0.0012*, 0.0004** in graph D.

E, F  Expression of CGAS-STING pathway genes in the RNA-seq data. The graph represents log2 RC ratios of [GSK-650394-treated/DMSO-treated control]. The molecules downregulated by the SGK1 inhibitor are marked with arrows in the CGAS-STING signal pathway schematized in F.

G  WB analysis demonstrating suppression of CGAS-STING signaling by the SGK1 inhibitor in VM-glial cultures. Representative blots are shown on the left. The levels of the active forms of CGAS-STING signaling molecules are quantified from 5–10 independent blots (right). Data are represented as mean ± SEM. One-way ANOVA. Significantly different at p = 0.001[#], 0.014* (CGAS), 0.00004[#], 0.0002* (STING), 0.017[#], 0.037* (pTBK1), 0.009[#], 0.045* (pIRF3), 0.012[#], 0.036* (pP65) in graph G.

H  Expression of *Ifnb* mRNA, a final product of the CGAS-STING pathway, determined by real-time PCR analyses. Data are represented as mean ± SEM. n = 3; Student's *t*-test. Significantly different at p = 0.0459[#], 0.0305* in graph H.

## SGK1 inhibition in glia ameliorates neuronal α-synuclein aggregation in *in vitro* α-synucleinopathic models

Given the observed anti-inflammatory and neurotrophic effects of glial SGK1 inhibition, we next examined whether SGK1 inhibition could be a useful therapeutic tool in PD. Intra-neuronal misfolding and the propagation of α-syn is a characteristic pathologic feature of PD and a target of PD therapeutic interventions. It is acknowledged that α-syn aggregation and propagation is promoted in a neuroinflammatory brain environment (reviewed in ref. Lema Tome et al, 2013). Specifically, recent studies have shown the ASC and caspase-1 released from microglia trigger the neuronal accumulation of α-syn aggregates (Wang et al, 2016; Gordon et al, 2018), and thus, inhibition of the glial inflammasome pathway abolished α-synucleinopathy in a PD animal model (Gordon et al, 2018). Because inflammation, especially the NLRP3-inflammasome pathway, was strongly inhibited by SGK1 inhibition (Fig 3A–D), we postulated that glial SGK1 inhibition might treat α-synucleinopathy and performed experiments to test that possibility in several *in vitro* α-synucleinopathy models. Because intra-neuronal SNCA (synuclein alpha, α-synuclein) aggregations are formed by increasing SNCA expression (Singleton et al, 2003; Chart ier-Harlin et al, 2004; Thakur et al, 2017), we prepared mDA neuronal cultures that overexpressed human SNCA and co-cultured them with VM-glia (schematized in Fig 5A). To facilitate SNCA aggregate formation, the co-cultures were treated with -SNCA preformed fibrils (PFFs), which act as seeds for intracellular SNCA aggregation in neuronal cells (Luk et al, 2009; Volpicelli-Daley et al, 2014). In this combined SNCA overexpression + PFF treatment condition, SNCA aggregates were readily formed and detected by immunoblot for SNCA and immunocytochemical analyses against phosphorylated SNCA at serine 129 (p129-SNCA) (Appendix Fig S3), which is associated with toxic SNCA fibril formation (Anderson et al, 2006). Treating the co-cultures with the SGK1 inhibitor GSK-650394 decreased the level of SNCA oligomers and monomers in a WB analysis (Fig 5B) and the p129-SNCA[+] cell counts (Fig 5C). However, the SGK1 inhibitor treatment effect was not manifested in primary neuron cultures without glial cells (Appendix Fig S4), suggesting that the SGK1 inhibitor effect to reduce α-synuclein pathology in the neuron + glia co-cultures was mediated by the inhibitor action on the glia. Synucleiopathy spreads mainly by neuron-to-neuron transfer of SNCA propagation (Luk et al, 2012). To further assess the effect of the SGK1

inhibitor on cell–cell SNCA transfer and aggregation, we used a bimolecular fluorescence complementation (Bi-FC) system (Bae et al, 2014), in which fluorescence emanates from the dimerization or oligomerization between SNCA fused to the amino (N) terminus (V1S) and that fused to the carboxy (C) terminus (SV2) fragment of Venus, a variant of yellow fluorescence protein (Fig 5D). Primary cultured VM-derived neural stem/precursor cells (NSCs) were individually transduced with the V1S and SV2 constructs and differentiated into neurons. Neither the neuronal cells expressing V1S nor those expressing SV2 produced Bi-FC fluorescence (data not shown). When the neuronal cells were co-cultured, however, fluorescence was visualized by the Bi-FC system (Fig 5E), validating the specificity of this model for assessing SNCA aggregation mediated by cell–cell SNCA transmission. As anticipated, the number of Bi-FC fluorescence puncta decreased greatly in the cultures treated with the SGK1 inhibitor GSK-650394 (Fig 5E).

Next, to test the effect of glial SGK1 knockdown on neuronal SNCA pathology, VM-glia were transduced with sh-SGK1 (or sh-control for the control), and medium was conditioned in the sh-SGK1-glia (sh-Cont-glia). The conditioned medium (CM) was administered to primary mDA neuron cultures (schematized in Fig 5 F). After 8–10 days of SNCA PFF treatment, the monomer and oligomer forms of SNCA were significantly lower in the mDA neurons treated with the sh-SGK1 glia-derived CM (sh-SGK1-GCM) vs. sh-cont-GCM, and the levels of nitrated SNCA, a pathologically modified SNCA (He et al, 2019), had decreased (Fig 5G). Furthermore, p129-SNCA immunoreactivity (% p129-SNCA + out of total TuJ1 + neurons) was significantly reduced by the administration of sh-SGK1-GCM (Fig 5H).

In addition to the NLRP3-mediated inflammatory environment, intra-neuronal oxidative and mitochondrial stresses are other prime causes of pathologic SNCA aggregation (Esteves et al, 2009; Li et al, 2011; Scudamore & Ciossek, 2018). Mitochondrial damage causes increased ROS through defective mitochondrial anti-oxidant functions and increased $Ca^{2+}$/calpain-mediated cleavage of SNCA through defective mitochondrial $Ca^{2+}$ buffering capacity, which ultimately results in oxidized/nitrated and truncated SNCA, respectively, which are prone to pathological aggregation (reviewed in ref. Esteves et al, 2011). Consistent with the observed glial neurotrophic and anti-inflammatory functions potentiated by SGK1 inhibition, intra-neuronal mitochondrial and oxidative stresses, estimated by MitoTimer, MitoSox, and DCF-staining, were greatly relieved in the

mDA neuronal cultures treated with GCM, and those effects in reducing neuronal oxidative/mitochondrial stress were enhanced by administering sh-SGK1-GCM (Appendix Fig S5A–C). Further

notably, the levels of reduced glutathione (GSH), a strong anti-oxidant, in the neuronal cells increased by GCM treatment, and the GCM effect was greater in the neuronal cultures treated with

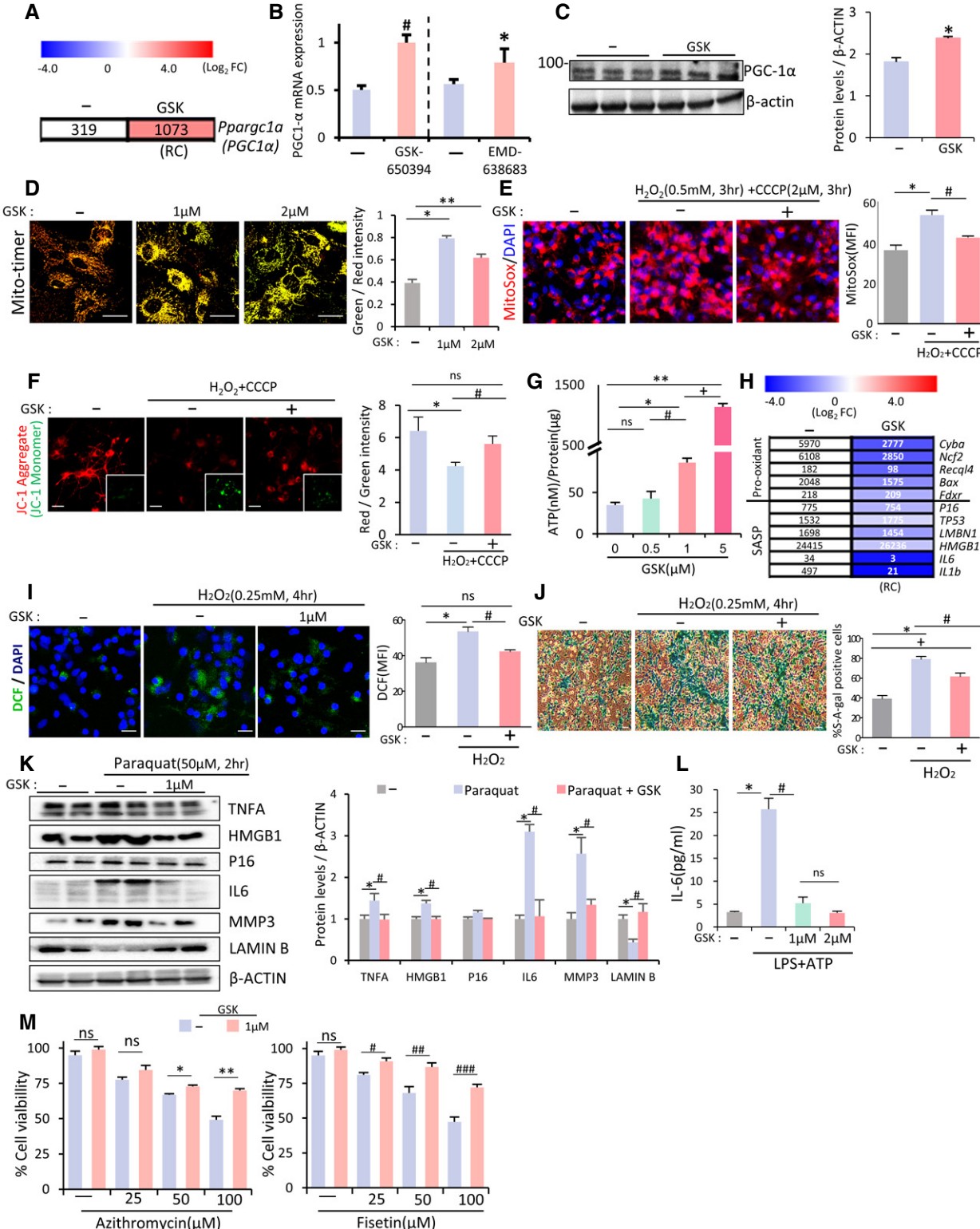

**Figure 4.**

**Figure 4. SGK1 inhibition protects glia from mitochondrial damage and cell senescence.**

A, B   *Ppargc1a* (PGC1α) mRNA expression upregulated in VM-glia treated with the SGK1 inhibitors in the RNA-seq data (A) and qPCR analysis (B). Data are represented
       as mean ± SEM. $n = 3$; Student's *t*-test. Significantly different at $P = 0.0011^{\#}$, 0.0499* in graph B.
C      Immunoblot for PGC1α protein expression. Data are represented as mean ± SEM. $n = 3$. Student's *t*-test. Significantly different at $P = 0.0077$* in graph C.
D      Mitochondrial biogenesis estimated by MitoTimer reporter gene expression in cultured VM-glia. Green fluorescence indicates newly synthesized mitochondria, and
       red fluorescence indicates old or damaged mitochondria. MFIs were estimated in three independent cultures (> 5 randomly selected microscopic fields/culture).
       Data are represented as mean ± SEM. One-way ANOVA. Significantly different at $P = 3.78E-08$*, 0.00003** in graph D.
E      Mitochondrial ROS levels estimated using the MitoSox red probe. Fluorescence intensity was measured using ImageJ, and ROS levels are presented as mean
       fluorescent intensity (MFI). Mitochondrial damage was induced by treatment with the mitochondrial toxins CCCP (2 μM) + $H_2O_2$ (500 μM) for 4 h. MFIs were
       estimated in three independent cultures (> 5 randomly selected microscopic fields/culture). Data are represented as mean ± SEM. One-way ANOVA. Significantly
       different at $P = 0.0013$*, $0.0273^{\#}$ in graph E.
F      Mitochondrial membrane potential (JC-1). The red fluorescence (JC-1 Aggregate) indicates disrupted mitochondrial membrane potential, and green fluorescence
       (JC-1 Monomer, insets) indicates intact potential. The inset images were taken from the same microscopic fields of the respective JC-1 aggregate images.
       Mitochondrial damage was induced by treatment with the mitochondrial toxins CCCP (2 μM) + $H_2O_2$ (500 μM) for 4 h. MFIs were estimated in three independent
       cultures (> 5 randomly selected microscopic fields/culture). Data are represented as mean ± SEM. One-way ANOVA. Significantly different at $P = 0.0006$*, $0.0093^{\#}$
       in graph F.
G      Intracellular levels of ATP, an indicator of mitochondrial oxidative phosphorylation. Data are expressed as mean ± SEM. $n = 3$ cultures; one-way ANOVA.
       Significantly different at $P = 0.0009$*, 0.00004**, $0.04913^{\#}$, $0.0024^{+}$ in graph G.
H–M    Glial cell senescence rescued by SGK1 inhibition. (H) Expression of genes associated with cell senescence in the RNA-seq data (VM-glia with vs. without GSK-
       650394 treatment). (I) Intracellular ROS levels estimated by DCF-DA. Data are represented as mean ± SEM. $n = 3$; one-way ANOVA. Significantly different at
       $P = 0.0074$*, $0.012^{\#}$ in graph I. (J) Senescence-associated β-galactosidase (SA-β-gal) activity. Data are represented as mean ± SEM. $n = 3$; one-way ANOVA.
       Significantly different at $P = 2.05E-06$*, $0.0042^{\#}$, $0.0006^{+}$ in graph J. (K) Immunoblots for proteins associated with cell senescence. Significantly different at
       $P = 0.008$*, $0.001^{\#}$ (TNFA), 0.01*, $0.009^{\#}$ (HMGB1), 7.32E-10*, $1.23E-8^{\#}$ (IL6), 9.62E-11*, $1.01E-5^{\#}$ (MMP3), 0.001*, $5.72E-4^{\#}$ (LAMIN B) in graph K. (L) IL-6 levels
       secreted from cultured VM-glia. The cytokine levels were estimated using ELISA of the media from the VM-glia with and without the SGK1 inhibitor. Data are
       represented as mean ± SEM. $n = 3$ cultures. One-way ANOVA. Significantly different at $P = 0.0028$*, $0.0062^{\#}$ in graph L. (M) Glial cell viability following treatment
       with senolytic compounds (azithromycin, fisetin). Data are represented as mean ± SEM. $n = 3$; Student's *t*-test. Significantly different at $P = 0.0073$*, 0.0018**,
       $0.0286^{\#}$, $0.0269^{\#\#}$, $0.0045^{\#\#\#}$ in graph M.

Data information: Scale bar, 20 μm.

shSGK1-GCM than sh-Cont-GCM (Appendix Fig S5D). These find-ings suggest that SGK1 inhibition in VM-glia potentiates glial actions that inhibit toxic SNCA aggregate formation at least in part by reliev-ing intra-neuronal oxidative and mitochondrial stresses.

## SGK1 inhibition in VM-glia protects mDA neurons from toxic insults

Next, we assessed the effects of glial SGK1 inhibition on *in vitro* cell survival and the resistance of mDA neurons, a neuronal type primarily affected in PD, to toxic insults. To this end, mDA neurons were co-cultured with glia (both derived from mouse VM) in which SGK1 was down- or upregulated (schematized in ref. Fig 6A). Cell death (TH+ cell counts) and neurite degeneration (TH+ fiber lengths) induced by extended culturing or exposure to the ROS producing agent $H_2O_2$ were greatly prevented in mDA neurons co-cultured with sh-SGK1-treated glia compared with those co-cultured with sh-control-treated glia (Fig 6B), whereas SGK1-overexpressing glia accelerated the mDA neurodegeneration especially induced by the toxin treatment (Fig 6C). Given that glia play their neuroprotec-tive/pro-inflammatory roles by secreting neurotrophic/pro-in-flammatory factors (reviewed in ref. Tang & Le, 2016; Du *et al*, 2017), the observed effects of SGK1 inhibition in glia are probably mediated, at least in part, in a paracrine manner. To test that suppo-sition, we conditioned the medium in glial cultures and added that conditioned medium (GCM) to primary mDA neurons (Fig 6D). Consistent with the neurotrophic nature of physiologic homeostatic glia, the cell viability of mDA neurons was promoted by adding GCM (sh-cont-GCM), and sh-Sgk1-GCM had stronger protective effects (Fig 6E). By contrast, when CM was prepared from glia pretreated with LPS and $H_2O_2$, treatment with the challenged GCM (Cll-GCM, Cll-sh-cont-GCM) worsened cell viability estimated by

TuJ1+ neuron counting and CCK8 cell viability assay (Fig 6E), indi-cating polarization of the glia into M1(A1)-type. The challenged GCM-mediated cell toxicity was much alleviated in mDA neuron cultures treated with GCM prepared from sh-Sgk1-transduced glia (Cll-sh-sgk1-GCM) (Fig 6E). It is acknowledged that normal glia and glia in the early stage of brain diseases (injuries) have neurotrophic properties, whereas those in the late (chronic) stage are detrimental. Thus, our findings indicate that SGK1 inhibition is a broadly appli-cable therapy regardless of the stage of neurodegenerative disorders (brain injuries) or the polarity of the glia because it can not only potentiate the neurotrophic functions of glia at the early disease stage, but also alleviate or correct the neurotoxic properties of glia in chronic disease environments. By contrast, the therapeutic utility of many drug candidates, such as those that simply target brain inflammation or glial phagocytosis, depends on the stage of the disease because inflammation and glial phagocytic activity are neuroprotective at the acute stage but detrimental at the chronic stage of CNS disorders.

Practically, pharmacologic inhibition is the most applicable ther-apeutic intervention. Chemical compounds, however, cannot be readily controlled to specifically act on glial cells without affecting neuronal cells in the brain being treated. Accordingly, if neuronal SGK1 inhibition has an opposite effect on neuronal cell viability (Chen *et al*, 2012), therapeutic efficacy might be eliminated or reduced in brains treated with a pharmacologic inhibitor. Indeed, we readily detected endogenous expression of SGK1 in neuronal cells (Appendix Fig S6A). To test the effect of SGK1 inhibition on neuronal cells, we conducted experiments with the SGK1 inhibitor in primary mDA neuronal cultures in which the glial population had been eliminated by Ara-C treatment (Kaech & Banker, 2006). Compared with the vehicle-treated control cultures, mDA neuron viability and resistance to a toxic H2O2 stimulus in cultures treated

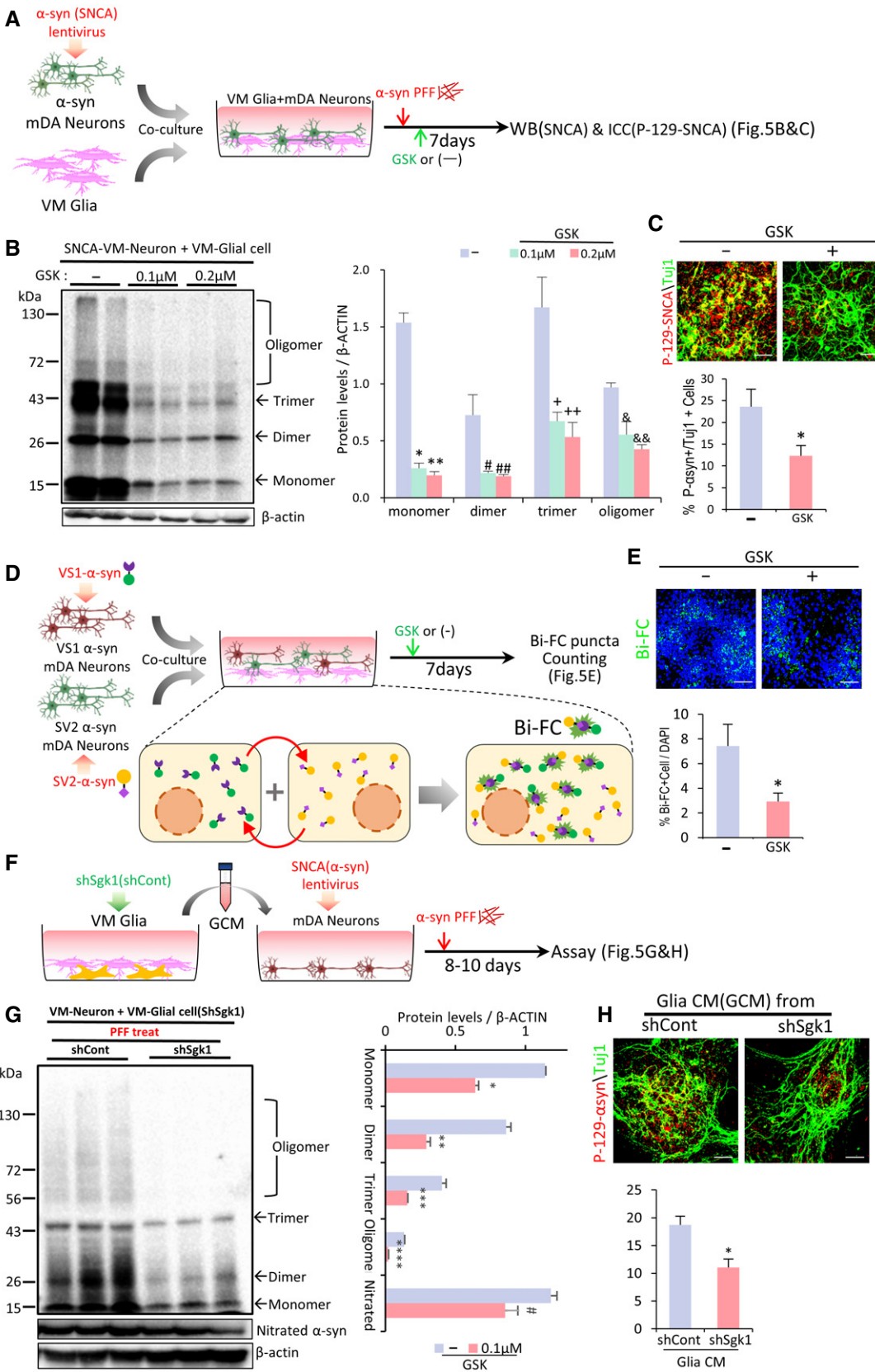

**Figure 5.**

**Figure 5. SGK1 inhibition potentiates the ability of VM-glia to inhibit neuronal SNCA aggregation and propagation *in vitro*.**

A   Schematic of the experimental procedure in (B and C).
B   Western blot analysis for detecting SNCA oligomers. Data are represented as mean ± SEM. *n* = 4; one-way ANOVA. Significantly different at $P$ = 2.5E-07*, 1.65E-07**, 0.022[#], 0.029[##], 0.046[+], 0.015[++], 0.0136[&], 0.0016[&&] in graph B.
C   Immunocytochemical detection of p-129-SNCA, associated with SNCA aggregation. Data are represented as mean ± SEM. *n* = 6. Significantly different at $P$ = 0.0363* in graph C.
D, E   Bi-FC-based assessment of the inter-neuronal propagation and aggregation of SNCA. (D) Schematic of the experimental procedures with Bi-FC system. Bi-FC[+] inclusions (puncta) were counted in (E). Data are represented as mean ± SEM. *n* = 5. Significantly different at $P$ = 0.0446* in graph E.
F–H   Glial SGK1 knockdown effect on neuronal SNCA pathology. (F), Schematic of the experimental procedure in (G, H). VM-glial cultures were transduced with sh-Sgk1 (or sh-cont) lentivirus. Medium was condition in sh-Sgk1-glia (or sh-cont-glia), and the conditioned medium (CM) treatment effects on neuronal SNCA aggregation were assessed by WB (G) and p-129-SNCA immunocytochemical (H) analyses. Data are represented as mean ± SEM. *n* = 3. Significantly different at $P$ = 0.00003*, 0.0003**, 0.0012***, 0.00004****, 0.0321[#] in graph G and $P$ = 0.0052* in graph H.

Data information: Student's *t*-test. Scale bar, 20 μm.

with the SGK1 inhibitor were slightly but insignificantly greater (Appendix Fig S6B), and the opposite effect was not observed in numerous independent experiments. Thus, SGK1 inhibition in neuronal cells is at least not harmful. Consistently, along with the neurotrophic roles of glia promoted by SGK1 inhibition, treating mixed mDA neuron-glia cultures with the SGK1 inhibitor always promoted the survival and resistance of mDA neurons (Fig 6F).

Next, we evaluated the contribution of each cell type of glia (microglia and astrocyte) and their interaction to the glial neuroprotective effects potentiated by SGK1 inhibition. To this end, we isolated astrocytes and microglia from the VM-glial cultures. CM was prepared from the cultured astrocytes (ACM) and microglia (MCM) and administered to cultured mDA neurons exposed to H2O2 (250 μM) (Appendix Fig S7A). Cell viability, estimated using the CCK8 assay, MAP2[+], and TH[+] neuronal counts, was improved in the cultures treated with either ACM or MCM, but ACM had a greater neuroprotective effect than MCM (Appendix Fig S7B, D, F). When astrocytes were pretreated with MCM and then CM was prepared from the MCM-pretreated astrocytes (M-ACM), the M-ACM treatment exerted a more potent neuroprotective effect than that seen with ACM treatment (Appendix Fig S7B, D, F), indicating that astrocytes positively interact with microglia in their neurotrophic functions. Expectedly, SGK1 knockdown (shSgk1 treatment) in cultured astrocytes and microglia potentiated the paracrine neurotrophic effects of those glial cell types (comparisons of sk-ACM vs. ACM and sk-MCM vs. MCM in Appendix Fig S7C, E, G). Furthermore, neuronal viability enhanced by treatments with skM-ACM and skA-MCM was significantly greater than those of M-ACM and A-MCM, respectively (Appendix Fig S7 C, E, G). These findings suggest that SGK1 inhibition in astrocyte and microglia potentiated paracrine neurotrophic effects not only from individual astrocytes and microglia, but also from the interplay between those glial cell types.

**Therapeutic potential of pharmacologic inhibition of SGK1 in PD mouse models**

Based on our *in vitro* data, we next sought to determine whether the pharmacologic inhibition of SGK1 could forestall the degeneration of mDA neurons and associated behavioral deficits in PD. To that end, we chose the MPTP mouse model of PD (Beal, 2001) using a sub-chronic systemic approach (MPTP intraperitoneal (i.p.) injection of 30 mg/kg for five consecutive days). Beginning one week after the initial MPTP injection series, the mice received i.p.

injections of the SGK1 inhibitor GSK-650394 (3 mg/kg) or DMSO (vehicle control) once a day. Behavioral assessments were performed every week by an independent experimenter blinded to treatment condition (schematized in Fig 7A). During the post-MPTP treatment period, the times required to accomplish the tasks in the beam and pole tests gradually increased in the MPTP-treated mice. The behavioral deficits were significantly alleviated in the mice treated with the SGK1 inhibitor beginning 4 weeks after the start of SGK1 treatment (5 weeks post-MPTP injection; Fig 7B and C). Furthermore, hypo-locomotor activity in the MPTP-PD mice was also significantly improved by treatment with the SGK1 inhibitor (Fig 7D). Compared with untreated mice (4,920 ± 469 cells, *n* = 6), injection of MPTP led to a massive loss of mDA neurons: TH[+] mDA neurons in the SN of the PD mice were reduced to 1,500 ± 164 cells (*n* = 20) 6 weeks post-MPTP injection. Treatment with the SGK1 inhibitor GSK-650394 significantly reduced mDA neuronal loss in the SN (2,570 ± 225 cells, *n* = 27, $P$ < 0.05) (Fig 7E and F). The TH[+] mDA neurons in the SNs of MPTP-treated mice had small cell bodies with blunted and fragmented neurites (Fig 7E enlarged images 1–2, G and H), indicating they were undergoing neurodegeneration. On the other hand, the mDA neurons in the SNs of MPTP-PD mice treated with the SGK1 inhibitor looked much healthier, with larger cell bodies and longer neurite outgrowths (Fig 7E enlarged images 3–4, G and H). Consistently, nigrostriatal dopaminergic innervation was also substantially protected by SGK1 inhibitor administration, as shown by the TH[+] fiber intensities in the striatum (Fig 7I and J). High-performance liquid chromatography–tandem mass spectrometry revealed that the SGK1 inhibitor compound was readily detectable in the brains of the MPTP-treated mice 30 min after administration (10.4 ± 1.0 ng/g brain, *n* = 3), indicating that the therapeutic effects might have been due to inhibitor action in brain glia.

We further assessed the therapeutic potential of the SGK1 inhibitor in another PD mouse model generated by injecting human SNCA PFF into the SN of the midbrain, in combination with adeno-associated virus (AAV)-mediated overexpression of human SNCA (Fig 7K), which is regarded as an animal model that is relevant to PD pathogenesis and disease progress (Thakur *et al*, 2017). Consistent with the previous findings observed in the MPTP-PD mouse model (Iwata *et al*, 2004; Stichel *et al*, 2005), SGK1 expression increased in the SNCA-injected SN and the increased SGK1 expression was ubiquitously detected in neuron, astrocyte, and microglia populations (Appendix Fig S8). Behavioral deficits, assessed by the beam and pole tests, progressed slowly in the α-synucleinopathic

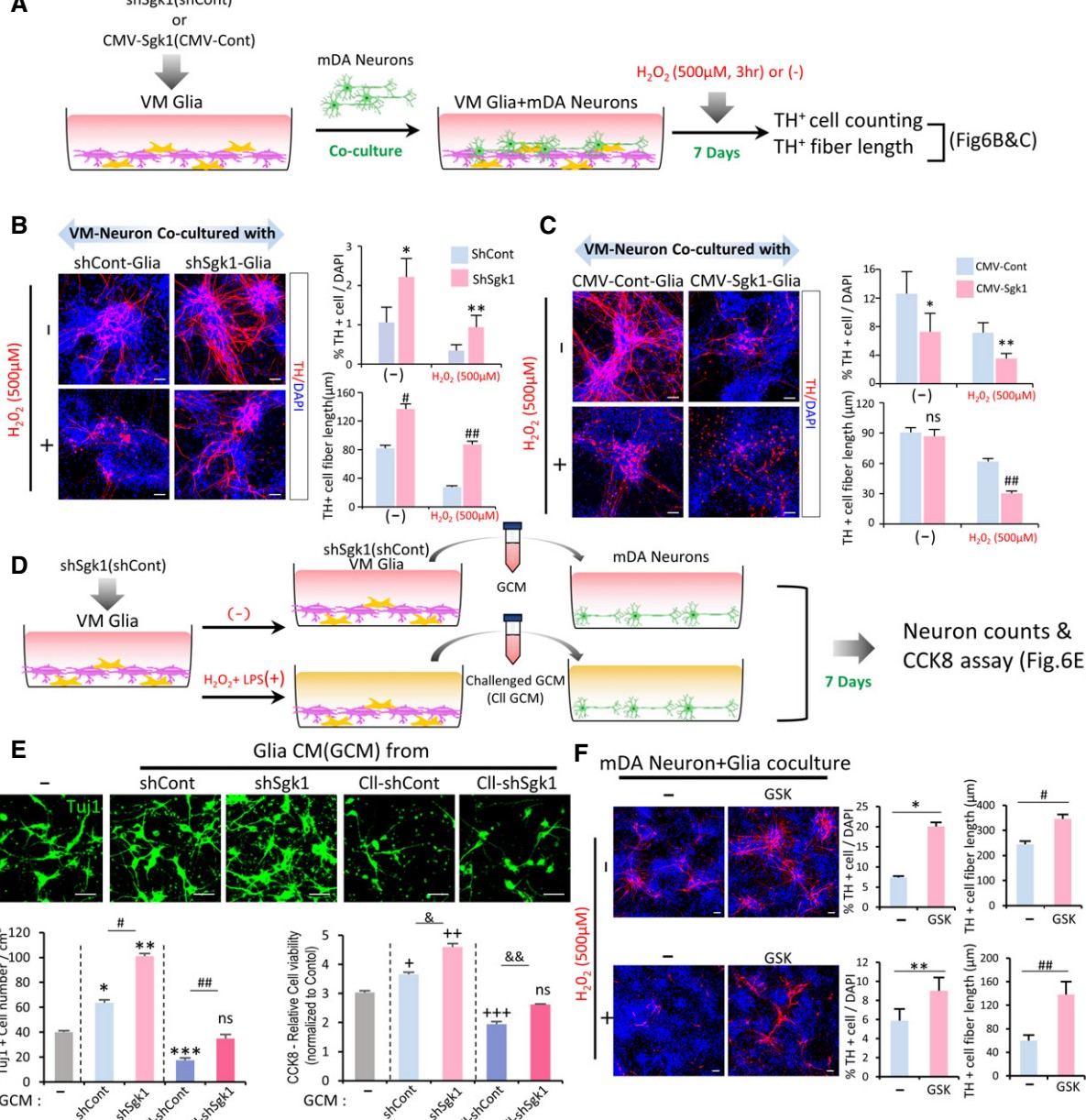

**Figure 6. SGK1 knockdown in VM-glia potentiates glial neuroprotective effects in VM-derived midbrain dopamine (mDA) neuron cultures.**

A–C Co-culture experiments. (A) Schematic for the experimental procedure. Glia derived from mouse VM were transduced with sh-Sgk1 (B) or SGK1 (CMV-Sgk1, C), and co-cultured with VM-derived mDA neurons. Seven days after co-culture, cells were exposed to $H_2O_2$ (500 μM, 3 h) or not (−), and then mDA neuron viability (TH+ cell counts) and neurite degeneration (TH+ fiber lengths) were estimated. Data are represented as mean ± SEM; $n = 4$ culture coverslips. Student's $t$-test. Significantly different at $P = 0.033$*, 0.0412**, 7.44E-23#, 1.39E-23## in graph B and $P = 0.0489$*, 0.0366**, 0.0078## in graph C.

D, E Glial neurotrophic functions potentiated by SGK1 downregulation were mediated in a paracrine manner. VM-glia transduced with sh-Sgk1 (or sh-cont as a control) were challenged with $H_2O_2$ + LPS for 3 h or not. Glial conditioned medium (GCM) was prepared in each glial culture condition and administered to mDA neurons primarily cultured from mouse VM (D). (E) Cell viability of the neuronal cells was estimated using CCK8 assays and TuJ1+ cell counts. Data are represented as mean ± SEM. $n = 3$ cultures; one-way ANOVA. Significantly different at $P = 0.0002$*, 3.29E-08**, 0.0003***, 3.66E-06#, 0.0029## (TuJ1+ cell counting), 0.0127+, 5.98E-06++, 0.0002+++, 0.0005&, 0.0078&& (CCK8 assays) in graph E.

F Effect of the SGK1 inhibitor treatment in the mixed mDA neuron + VM-glia co-cultures. Data are represented as mean ± SEM. $n = 3$. Student's $t$-test. Significantly different at $P = 0.0068$*, 0.0368**, 0.0249#, 1.48E-27## in graph F.

Data information: Scale bar, 20 μm.

PD mice. SGK1 inhibitor treatment significantly alleviated the progress of the motor symptoms from 10 to 12 weeks (Fig 7L–N). As previously described (Thakur *et al*, 2017), there is substantial immunoreactivity for p129-SNCA in the SNs of mice injected with the combined SNCA PFF + SNCA-AAV2, indicative of pathologic SNCA aggregation. In the SNs of SNCA-PD mice treated with the inhibitor GSK-650394, p129-SNCA immunoreactivity was significantly reduced, especially in TH[+] DA neurons (Fig 7O and P). Along with the reduction in SNCA pathology, the degeneration of SN DA neurons and axons (Fig 7O, Q–S) and loss of striatum TH[+] fiber intensities (Fig 7T and U) was significantly blunted in the animals treated with the SGK1 inhibitor.

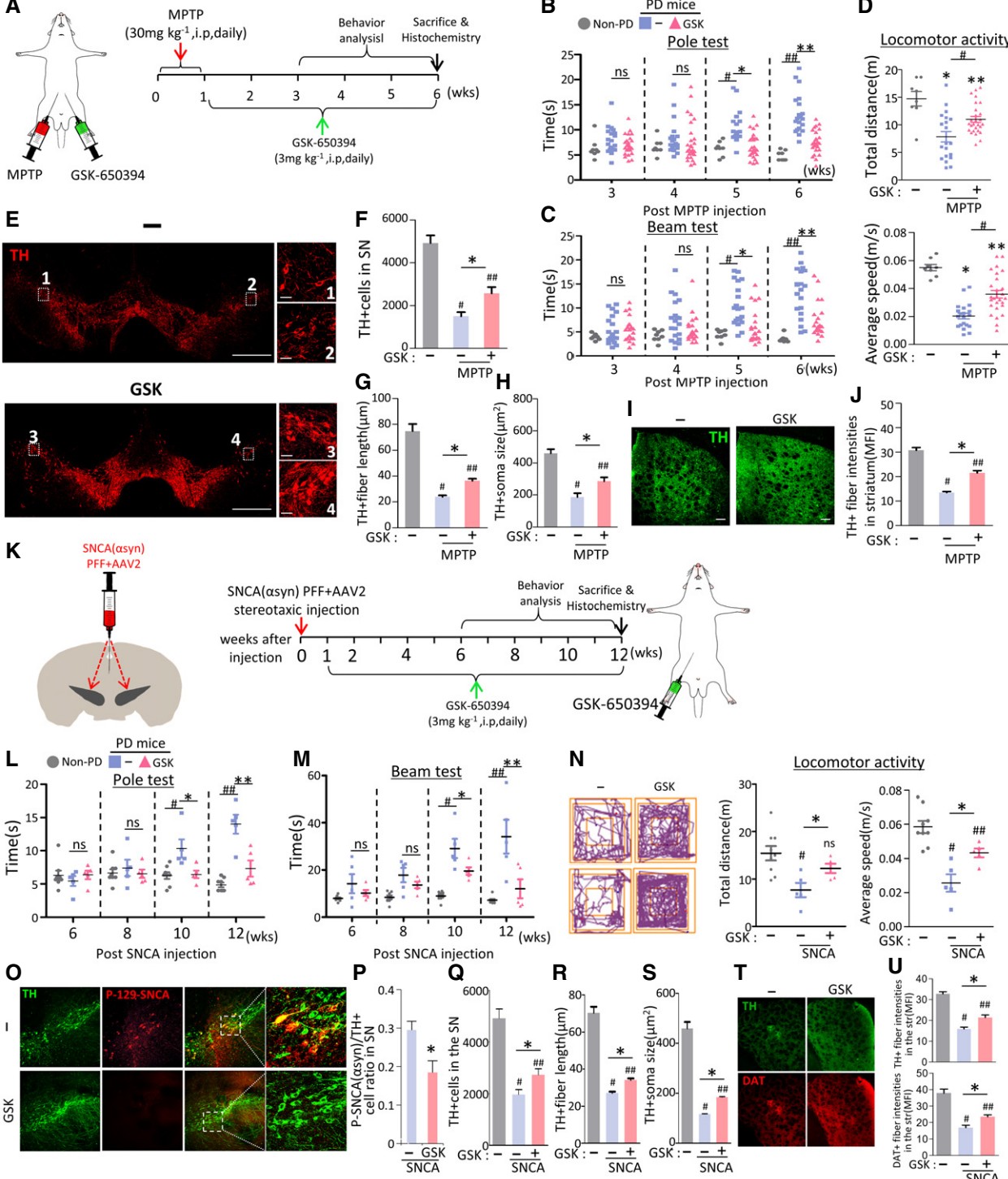

**Figure 7.**

◀

**Figure 7. Therapeutic effects of SGK1 inhibitor treatment in PD mouse models.**

A–J Effect of treatment with the SGK1 inhibitor GSK-650394 in the MPTP-PD mouse model. (A) Overview of the experimental procedure. (B–D) Behaviors of MPTP-PD mice assessed by pole (B), beam (C) at 3–6 weeks post-MPTP injection, and locomotor activity (total distance & average speed, D) tests at 6 weeks post-MPTP injection. $n = 20$ GSK-650394-treated MPTP-PD mice, 14 vehicle-treated MPTP-PD mice, and 8 MPTP-untreated non-PD mice. Data are represented as mean ± SEM. Significantly different at $P = 0.0012^{\#}$, $0.0001^*$, $3.09E\text{-}08^{\#\#}$, $8.21E\text{-}08^{**}$ in graph B and $P = 0.0009^{\#}$, $0.0035^*$, $1.03E\text{-}06^{\#\#}$, $0.00003^{**}$ in graph C and $P = 0.0001^*$, $0.0284^{**}$, $0.0101^{\#}$ (total distance), $1.49E\text{-}08^*$, $0.0005^{**}$, $0.0001^{\#}$ (average speed) in graph D. Histologic assessment of the number of TH$^+$ mDA neurons (E, F), neurite lengths (E, G), and cell body sizes (E, H) of the mDA neurons in the SN and TH$^+$ fiber intensities in the striatum (I, J). Shown in (E and I) are the representative images for TH$^+$ DA neurons in the SN and TH-immunoreactivity in the striatum, respectively. TH$^+$ DA neurons in boxed areas of (E) are enlarged in right. Data are represented as mean ± SEM. $n = 3$–4 mice per group. Significantly different at $P = 7.09E\text{-}09^{\#}$, $0.0001^{\#\#}$, $0.0162^*$ in graph F and $P = 5.09E\text{-}09^{\#}$, $5.10E\text{-}09^{\#\#}$, $6.66E\text{-}09^*$ in graph G and $P = 4.89E\text{-}08^{\#}$, $0.0001^{\#\#}$, $0.0152^*$ in graph H and $P = 5.10E\text{-}09^{\#}$, $0.0128^{\#\#}$, $0.0001^*$ in graph J.

K–U Therapeutic effects of GSK-650394 in SNCA-PD model mice. Behaviors (L–N) and mDA neuron degeneration in the SN (O–U) of the SNCA-PD mice were assessed at 6–12 (L, M) and 12 weeks post-SNCA injection before (N) and after sacrifice (O–U). Significantly different at $P = 0.008^{\#}$, $0.021^*$, $0.00001^{\#\#}$, $0.0009^{**}$ in graph L and $P = 0.00002^*$, $0.0344^*$, $0.0005^{\#\#}$, $0.0067^{**}$ in graph M and $P = 0.0057^{\#}$, $0.0345^*$ (total distance), $0.0001^{\#}$, $0.0375^{\#\#}$, $0.0338^*$ (average speed) in graph N. (P) SNCA pathology assessed by p129-α-syn immunoreactivity in TH + mDA neurons in the SN. Significantly different at $P = 0.019^*$ in graph P and $P = 0.0001^{\#}$, $0.0009^{\#\#}$, $0.032^*$ in graph Q and $P = 5.1E\text{-}09^{\#}$, $5.1E\text{-}09^{\#\#}$, $0.0001^*$ in graph R and $P = 5.1E\text{-}09^{\#}$, $5.1E\text{-}09^{\#\#}$, $0.0005^*$ in graph S and $P = 5.1E\text{-}09^{\#}$, $7.83E\text{-}09^{\#\#}$, $0.0025^*$ (TH), $0.0006^{\#}$, $0.0062^{\#\#}$, $0.041^*$ (DAT) in graph U. $n = 5$ GSK-650394-treated SNCA-PD mice, 5 vehicle-treated SNCA-PD mice, and 8 SNCA-untreated non-PD mice.

Data information: Data are represented as mean ± SEM. One-way ANOVA. Scale bar, 100 μm.

## SGK1 silencing using intra-SN injection of sh-SGK1 AAV9 is another potential therapeutic tool for PD

Our final experiment tested whether silencing SGK1 using sh-RNA delivery could be an effective therapeutic tool for PD. Before testing the therapeutic effect in a PD mouse model, we examined the efficiency of gene delivery using the AAV vector, which has been suggested as a desirable therapeutic vector for CNS disorders (reviewed in ref. Hudry & Vandenberghe, 2019). AAV serotype 2 (AAV2), the most well characterized and frequently used in clinical trials, has strong neurotropism without transgene expression in glial cells (McCown, 2011; Oh et al, 2015), which excluded it from this study because we wanted to deliver genes efficiently to glial cells. We tested AAV9-mediated gene delivery because it was also suggested as a compelling candidate for gene therapy in CNS disorders (Pattali et al, 2019). We carried out stereotaxic injection of AAV9 carrying the reporter green fluorescent protein (GFP) gene under the control of the universal CMV promoter (GFP-AAV9) into the SN. The GFP exogene expression was detected to cover whole midbrain regions and was readily detected in astrocytes and microglia: Of the GFP$^+$ cells, GFAP$^+$ astrocytes accounted for 40.1 ± 6.8%, Iba1$^+$ microglia for 19.8 ± 4.0%, and NeuN$^+$ neurons for 34.3 ± 5.5% (Fig 8A and B). Given that efficient gene transfer to glial cells, we constructed AAV9 expressing sh-SGK1 to test its therapeutic effects (Fig 8C). SGK1 mRNA expression was efficiently downregulated in SNCA model PD mice by intra-SN injection with sh-Sgk1-AAV9 (Fig 8D). Compared to the sh-control-treated mice, immunohistochemical analysis against SGK1 revealed 50, 60, and 45% reduction in SGK1-immunoreactive cells in the neuron, astrocyte, and microglial populations of the SN treated with sh-Sgk1-AAV9, respectively (Fig 8E). In addition to preventing SNCA pathology (% p129-SNCA$^+$ cells), mDA neuron degeneration in the SN (Fig 8I–M) and loss of TH$^+$ and DA transporter (DAT)$^+$ fiber intensities in the striatum (Fig 8N and O), intra-SN injection of sh-Sgk1-AAV9 ameliorated behavioral deficits in SNCA model PD mice (Fig 8F–H).

## Suppression of inflammation and cell senescence in the SN of PD mice underlies the therapeutic effects attained by SGK1 inhibition

Compared with the vehicle-treated control PD mice, Iba1$^+$ microglia accumulated less abundantly in the SN and neighboring areas of the GSK-650394-treated SNCA-PD mice (Fig 9A and B). In morphometric measurements, inflated (hypertrophic) microglia, a characteristic of inflammatory reactive microglia, were much ameliorated in the SNs of mice treated with the SGK1 inhibitor (see Insets in Fig 9A and H), with a significant reduction in microglial soma sizes (Fig 9 C). Furthermore, the expression of the pro-inflammatory M1 marker CD16/32 was localized in the lower proportion of microglia in the SNs of mice treated with the inhibitor (Fig 9D and E). In accord with the in vitro findings shown in Figs 1J–M and 2D–E, mRNA expression of the pro-inflammatory cytokines (Il1b, Tnfa) and key inflammasome components (Nlrp3, Asc, Casp1) was significantly lowered in the SNs of MPTP- and SNCA-PD model mice by administering the SGK1 inhibitor (Fig 9F and G). The key NLRP3-inflammasome component ASC was barely detected in un-lesioned mice (data not shown and (Gordon et al, 2018)), whereas ASC immunoreactivity was abundantly detected and localized in hypertrophic Iba1$^+$ microglia in the SN area of control SNCA-PD mice (Fig 9H). Consistent with the suppression of NLRP3 inflammasome activation with glial SGK1 inhibition (Fig 3A–D), ASC immunoreactivity was greatly reduced in the animals treated with the SGK1 inhibitor (Fig 9H–J).

SA-β-Gal activity was detectable in the midbrains 12 weeks after SNCA-PFF + AAV injection (but not in the control and MPTP-PD mice). Consistent with the in vitro observation that glial cell senescence was reduced by SGK1 inhibition (Fig 4H–M), SA-β-gal staining intensity in the mouse midbrain was markedly lowered by inhibitor administration (Fig 9K and L). Next, to determine the cell type identity of the senescent cells, we attempted to use co-immunofluorescent staining for cell senescence markers (p16, TBP) with neuron- or glial-specific markers, but we failed to get specific and positive staining for the senescence markers. Thus, although the cell type undergoing changes in cell senescence in the PD model remains to be identified, our findings collectively suggest that SGK1 inhibition could be used to treat PD by suppressing the glial activation of inflammation, especially the NLRP3-inflammasome, and cellular senescence (Appendix Fig S9).

# Discussion

For decades, neuronal cells have been the central cell type targeted for drug development in CNS disorders. However,

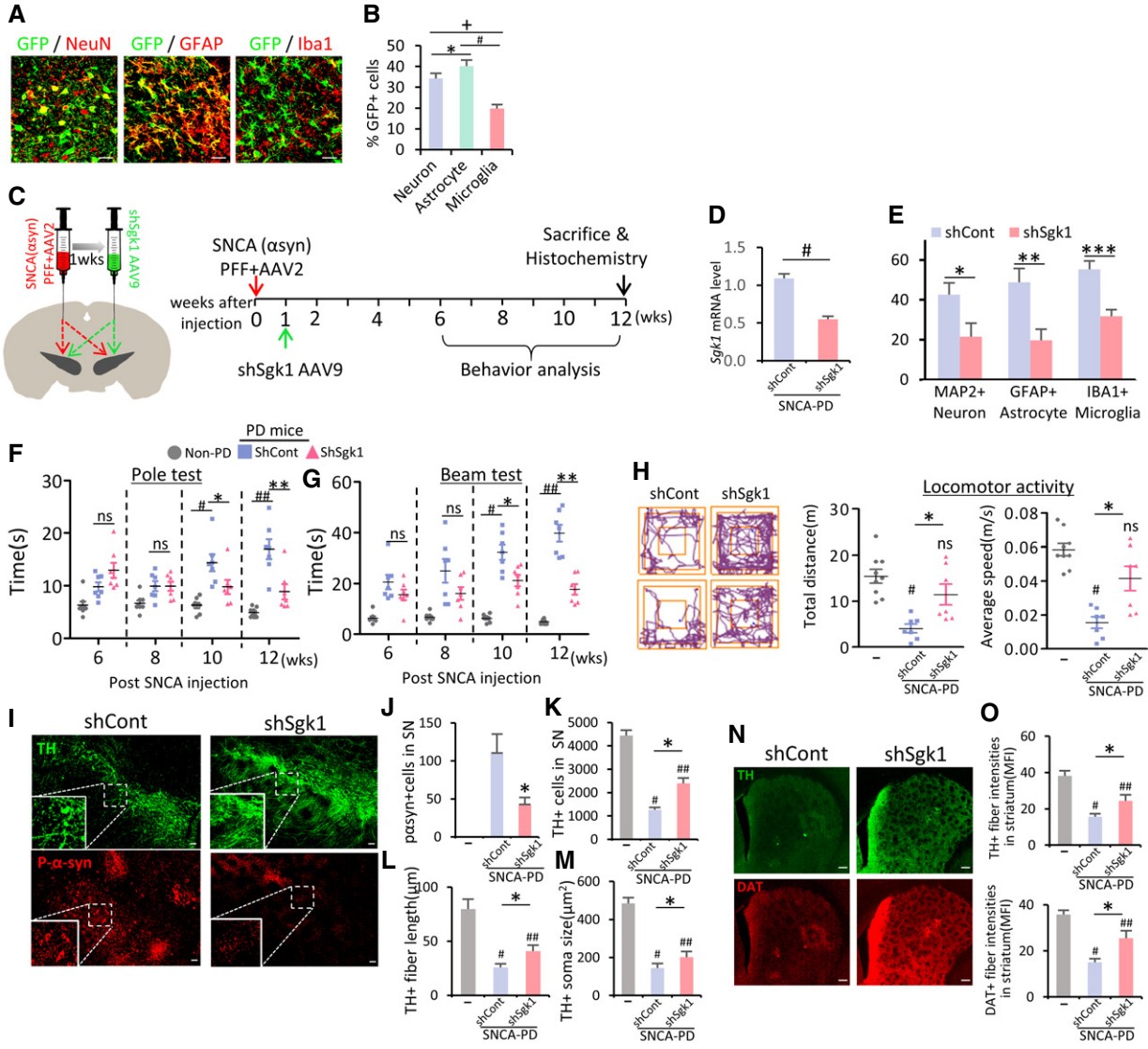

**Figure 8. Intra-SN injection of sh-Sgk1 AAV9 as another potential therapeutic strategy in PD.**

A, B  Efficiency of AAV9-mediated gene delivery. AAV9 viruses expressing the reporter GFP gene were stereotaxically injected into the SN of mice, and GFP expression in astrocytes, microglia, and neuronal cells was quantified. Data are represented as mean ± SEM. $n = 3$ mice (> 5 randomly selected microscopic fields per animal were counted). Significantly different at $P = 0.006^*$, $5.1E-09^\#$, $5.36E-09^+$ in graph B.

C–O  Therapeutic effects of intra-SN injection of sh-Sgk1 AAV9 in SNCA-PD mouse model. (C) Schematic of the experimental procedure. (D and E) SGK1 knockdown effect *in vivo* by treatment with shSgk1-AAV9. Total SGK1 mRNA levels were estimated in the midbrain SN of the PD mice injected with shSgk1 (sh-Control as control) AAV9 using real-time PCR (D). Cell type-specific SGK1 knockdown was estimated by counting SGK1-immuonoreactivity in neuron (MAP2[+]), astrocyte (GFAP[+]), and microglia (Iba1[+]) populations (E). Data are represented as mean ± SEM. $n = 4$ mice in each group. Significantly different at $P = 0.0004^\#$ in graph D and $P = 0.049^*$, $0.007^{**}$, $0.002^{***}$ in graph E. (F–H) Behavior of the PD mice assessed by pole (F), beam (G) at 6–12 weeks, and locomotor activity (H) tests at 12 weeks post-SNCA injection. Significantly different at $P = 0.0003^\#$, $0.0428^*$, $7.46E-06^{\#\#}$, $0.0012^{**}$ in graph F and $P = 8.04E-08^\#$, $0.004^*$, $1.62E-09^{\#\#}$, $3.3E-06^{**}$ in graph G and $P = 0.0002^\#$, $0.0194^*$ (total distance), $9.54E-06^\#$, $0.0047^*$ (average speed) in graph H. (I–O) Histologic assessment of p129-SNCA immunoreactivity (I, J), TH[+] mDA neuron counts (I, K), neurite lengths (L), and cell body sizes (M) in the SN and TH[+] and DA transporter (DAT) fiber intensities in the striatum (N, O). Significantly different at $P = 0.0001^*$ in graph J, $P = 2.35E-25^\#$, $1.09E-16^{\#\#}$, $4.36E-08^*$ in graph K and $P = 1.5E-40^\#$, $1.08E-24^{\#\#}$, $2.32E-07^*$ in graph L and $P = 9.07E-20^\#$, $2.59E-16^{\#\#}$, $0.05^*$ in graph M and $P = 4.62E-22^\#$, $5.11E-12^{\#\#}$, $3.49E-06^*$ (TH), $1.58E-22^\#$, $8.13E-09^{\#\#}$, $7.42E-09^*$ in graph O. $n = 7$ sh-SGK1-AAV9-treated PD mice, 7 sh-cont-AAV9-treated PD mice, and 8 SNCA-untreated mice. Data are represented as mean ± SEM. One-way ANOVA. Scale bar, 100 µm.

effective disease-modifying therapies that rescue degenerating neurons have not yet been developed. The idea of correcting pathologic brain environments by targeting glial cells has been appearing on the therapeutic horizon for CNS disorders (Hamby & Sofroniew, 2010). This study provides an example for how to

develop an effective glial targeting strategy to treat neurodegenerative disorders, especially PD.

We initiated this study by identifying SGK1 as a molecule that is downregulated at the transcriptional level by Nurr1 + Foxa2 and thus mediates Nurr1 + Foxa2-induced anti-inflammatory functions

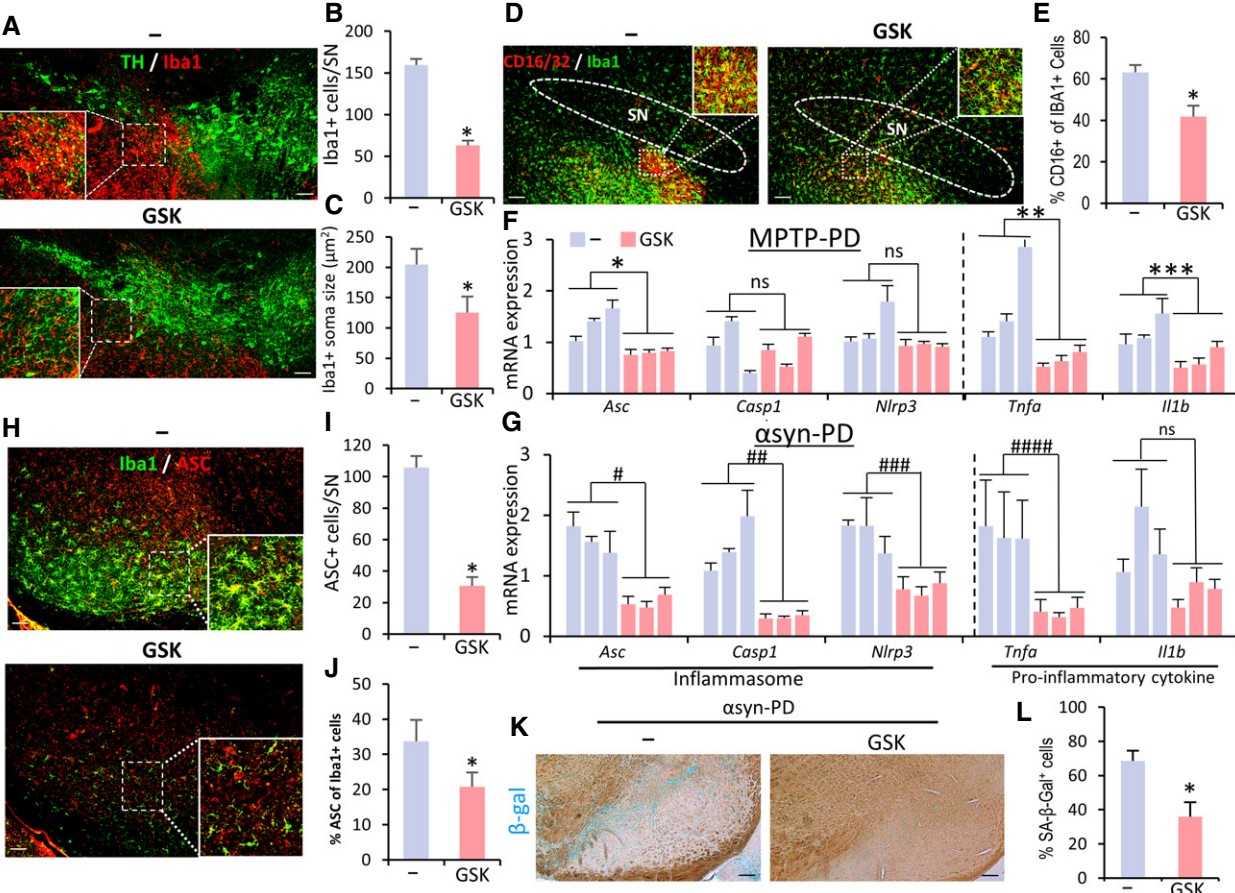

**Figure 9. The Sgk1 inhibitor treatment blunted microglial activation, NLRP3-inflammation, and cell senescence in the SNs of PD mice.**

A–C  Quantitative and morphometric analyses of microglia in the SN area of SNCA-PD mice. Shown in (A) are representative views of Iba1[+] microglia neighboring TH[+] mDA neurons in the SNs of PD mice treated with GSK-650394 and vehicle. Insets, enlarged views for the boxed areas. Data are represented as mean ± SEM. n = 3 mice per group. Significantly different at P = 1.49E-11* in graph B and P = 0.000003* in graph C.

D, E  The M1 marker CD16/32 expression in Iba1[+] microglia. Data are represented as mean ± SEM. n = 3 mice per group. Significantly different at P = 0.0016* in graph E.

F, G  mRNA expression of NLRP3-inflammasome components and pro-inflammatory cytokines in MPTP (F) and SNCA (G) PD mice injected with the SGK1 inhibitor or vehicle (control). Each bar represents the mean ± SEM of 3 PCR values from each animal, n = 3 mice per group. Significantly different at P = 0.0372*, 0.0173**, 0.0003*** in graph F and P = 0.002#, 0.0113##, 0.0053###, 0.0001#### in graph G.

H–J  ASC expression in the SNs and SN microglia of SNCA-PD mice. n = 5 mice per group. Data are represented as mean ± SEM. Significantly different at P = 1.91E-09* in graph I and P = 0.0079* in graph J.

K, L  β-gal-stained cell senescence in SNCA-PD mouse SN. Data are represented as mean ± SEM. n = 5 mice per group. Significantly different at P = 0.0001*.

Data information: Student's t-test. Scale bar, 100 μm.

in glia. How Nurr1 + Foxa2 regulates SGK1 expression is intriguing but has not been identified in this study. Nurr1 and Foxa2 have naïve roles as transcription factors that activate gene transcription. Thus, Nurr1 + Foxa2-induced downregulation of SGK1 is likely mediated in an indirect manner via activation of molecular transcriptions or pathways that can inhibit SGK1 transcription. SGK1 transcription is controlled by a wide variety of stimuli including cell stress, hormones, cytokines, cytosolic $Ca^{2+}$ activity, and a number of intracellular pathways. Specifically, oxidative stress and DNA damage are common stimulators of SGK1 transcription (Leong et al, 2003; Lang et al, 2006), while Nurr1 and Foxa2 have been shown to ameliorate those cellular stresses by promoting trophic and stress responses, mitochondrial gene expressions, and DNA repair

function (Sousa et al, 2007; Song et al, 2009; Malewicz et al, 2011; Kadkhodaei et al, 2013; Oh et al, 2015), implying that SGK1 expression was indirectly controlled by Nurr1 and Foxa2 actions to relieve cellular stresses. In addition, Nurr1 has been shown to exert an epigenetic repressor function by forming a repressor complex containing corepressor element 1 silencing transcription factor (CoREST) complex (Saijo et al, 2009). Thus, a Nurr1 (Foxa2)-mediated action to directly inhibit SGK1 gene transcription could not be excluded.

We found that gene knockdown and pharmacological inhibition of SGK1 in glial cells potently suppressed glial inflammatory reactions mediated by the NFκB, inflammasome, and STING intracellular pathways. Furthermore, glial mitochondrial oxidative stress and

cell senescence were also relieved by SGK1 inhibition, whereas glial capacity to cleanse glutamate toxicity was potentiated. Ultimately, silencing and pharmacological inhibition of SGK1 potentiated the neurotrophic roles of glia and rescued mDA neurons from degeneration and α-synucleinopathy in multiple *in vitro* and *in vivo* PD models. It is likely that the multifaceted therapeutic functions promoted by SGK1 inhibition are linked together by multiple positive feedback regulatory loops: SGK1 inhibition in glial cells first blocks the inflammatory pathways by inhibiting the intracellular NFκB pathway, which blocks the NFκB-mediated expression of pro-inflammatory cytokines and inflammasome components. At the same time, downregulation of NFκB promotes glial GLT1 (SLC1A2) expression (Fine *et al*, 1996; Su *et al*, 2003; Boycott *et al*, 2008; Jiang *et al*, 2019) and activity to clear glutamate-mediated excitatory toxicity. Reduced glial secretion of pro-inflammatory cytokines (TNFα) could decrease intracellular oxidative and mitochondrial stress (Blaser *et al*, 2016) in neighboring glia. Cell senescence in the glia is thus ameliorated by the lowered ROS levels. At the same time, the inflammasome and STING pathways are deactivated in the glia treated with the SGK1 inhibitors and sh-Sgk1, not only by downregulated expression of the key components of the pathways, but also by relief from mitochondrial stress, which is closely linked with the activation of those pathways (Sorbara & Girardin, 2011; Heid *et al*, 2013; Liu *et al*, 2016; Panicker *et al*, 2019). On the other hand, oxidative and mitochondrial stress in neuronal cells are also relieved in the same manner by glia treated with SGK1 inhibition, and another positive feedback regulation could also be established among oxidative and mitochondrial stress and abnormal SNCA aggregation: lowered ROS and mitochondrial dysfunction could reduce SNCA aggregate formation and propagation (Lee, 2003; Esteves *et al*, 2009; Esteves *et al*, 2011; Li *et al*, 2011; Scudamore & Ciossek, 2018). Reciprocally, with relief from α-synucleinopathy, mitochondria become healthier and more functional, and intraneuronal ROS is reduced (Hsu *et al*, 2000; Hashimoto *et al*, 2003; Musgrove *et al*, 2019). The therapeutic effects attained by glial SGK1 inhibition are summarized in Appendix Fig S9.

In addition to the knockdown strategy using sh-RNA delivery, systemic administration of the SGK1 inhibitor GSK-650394 has been demonstrated in this study as a potential therapeutic tool to treat PD. Detectable levels of the inhibitor were found in the brains of PD mice, suggesting that the therapeutic effects seen in the PD mice after systemic injection of the SGK1 inhibitor occurred because the inhibitor permeated into the brain and acted on the glia there. However, the inhibitor concentration detected in the brains of PD mice after i.p. injection ($10.4 \pm 1.0$ ng/g brain) was low compared with the plasma concentration ($342.3 \pm 146.3$ ng/ml), resulting in a brain to plasma concentration ratio of 0.03. More extensive optimization studies are required to generate more efficient SGK1 inhibitor derivatives to improve the brain pharmacokinetics *in vivo*. On the other hand, PD is regarded as an auto-immune disease, with evidence for the infiltration of peripheral T lymphocytes in postmortem brain tissues of PD patients and alterations in T-cell subsets in the blood of PD patients (Bas *et al*, 2001; Baba *et al*, 2005; Brochard *et al*, 2009; Sie *et al*, 2014; Fahmy *et al*, 2019). Specifically, recent experimental evidence has shown the pathologic roles of helper T17 (Th17) lymphocytes in PD (Sommer *et al*, 2019), whereas the adoptive transfer of regulatory T (Treg) cells has

manifested neuroprotective functions in an MPTP murine model of PD (Reynolds *et al*, 2007), and altered Treg cells have shown suppressive functions in PD patients (Saunders *et al*, 2012). Notably, SGK1 has been shown to promote the activation and differentiation of Th17 lymphocytes at the expense of those of Treg cells (Du *et al*, 2018; Spagnuolo *et al*, 2018; Wu *et al*, 2018). Thus, portions of the therapeutic effects seen following the systemic administration of the SGK1 inhibitor could be mediated through the correction of the Th17/Treg cell imbalance in the PD animals. Studies are needed to address and identify the therapeutic effects of SGK1 inhibition attained through T-cell modulation in PD models.

In conclusion, our results provide evidence that SGK1 inhibition in astrocytes and microglia can be therapeutically used in PD. Because the glial pathologies shown to be improved by SGK1 inhibition in this study commonly underlie other CNS disorders, SGK1 inhibition might eventually become a therapeutic intervention that is broadly applicable to a variety of neurodegenerative disorders and neurologic injuries.

## Materials and Methods

### Cell cultures and treatments

#### Primary mixed glial culture
Mixed glia were isolated from mouse or rat VMs on postnatal day 1 (P1)–P5, using a previously described protocol (Oh *et al*, 2015). In brief, VMs were removed, triturated in DMEM (Thermo Scientific Inc., Waltham, MA, USA) containing 10% FBS (HyClone, GE Healthcare Life Sciences, Pittsburgh, PA, USA), and plated in 75-cm$^2$ T-flasks. When cell confluence reached 80–90%, cells were harvested with 0.125% trypsin and sub-cultured on a culture surface coated with poly-d-lysine (Millipore Sigma, Louis, MO, USA).

#### Primary astrocyte and microglial culture
Microglia and astrocytes were isolated from mixed primary glial cultures obtained from the VMs of a P3 mouse or rat. Briefly, VMs were dissected in Ca$^{2+}$- and Mg$^{2+}$-free HBSS (Thermo Scientific Inc.) and incubated in a 0.125% trypsin solution for 10 min at 37 °C. The resulting cell suspensions were diluted in complete medium (DMEM-F12, Thermo Scientific Inc.) with 10% FBS, 10% horse serum (HyClone), 1 mM L-glutamine (Life Technologies, Carlsbad, CA, USA), and 1% PenStrep (Thermo Scientific Inc.) and cultured for 14 days. Floating microglia and attached pure astrocytes were collected from the mixed glial cultures by shaking the flask on an orbital shaker at 0.4 g for 12 h at 37°C.

#### Cultures enriched with mDA neurons
Neural stem/precursor cells cultured from VM in the early embryonic days have mDA neurogenic potential without glial differentiation (Studer *et al*, 1998). Therefore, mDA neuron-enriched cultures were attained by differentiating NSCs cultured from the VMs of mouse (ICR) embryos at E10.5 or VMs of rat (Sprague Dawley) embryos at E12, as previously described (Song *et al*, 2018). In addition, mDA neurons were directly isolated and cultured from rat VMs by adopting the method for conventional primary neuron culture (Kaech & Banker, 2006). Briefly, VM tissues from rats at E14–16

were triturated to single cells with papain (Sigma) and plated on culture dishes pre-coated with polyornithine (PLO)/fibronectin (FN) in neurobasal medium supplemented with B27 and L-glutamine (all from Thermo Scientific Inc.). Glial cell populations were eliminated by adding Ara-C (3 μM; Sigma) for days 3–5.

### mDA neuron-VM-glia co-culture

The differentiated VM-NSC culture (mDA neuron-enriched culture) was harvested and mixed with primary culture for VM-glia in the ratio 2:1 and then cultured. For analyses, the cells were treated with SGK1 inhibitors [GSK650394 (1–2 μM) (Tocris Bioscience, Bristol, U. K.) or EMD638683 (1–2 μM) (Chemscene, Monmouth Junction, NJ, USA)] with or without toxins, LPS (0.25 μg/ml, Sigma), $H_2O_2$ (150–500 μM, Sigma), ATP (2.5 mM, Sigma), or Paraquat (50 μM, Sigma). The specificity of the SGK1 inhibitors has been shown with low levels of cross-reactivity with other members of the AGC Ser/Thr protein kinase family such as AKT or PDK1(Sherk *et al*, 2008; Ackermann *et al*, 2011).

### *In vitro* overexpression of Nurr1, Foxa2, SGK, and knockdown of SGK1

Lentiviral vectors expressing Nurr1 (mouse), Foxa2 (rat), and SGK1 (mouse), all from mouse genes under the control of the CMV promoter, were generated by inserting the respective cDNA into the multi-cloning site of pCDH (System Biosciences, Mountain View, CA, USA). The lentiviral vector pGIPZ-shSGK1 was purchased from Open Biosystems (Huntsville, AL, USA). The empty backbone vector (pCDH or pGIPZ) was used as the negative control. Titers of the lentiviruses were determined using a QuickTiter HIV Lentivirus Quantitation Kit (Cell Biolabs, San Diego, CA, USA), and 20 μl each of the viruses with $1.0 \times 10^6$ transducing units (TU)/ml were mixed with 2 ml of medium and added to $1–1.5 \times 10^6$ cells/6 cm-dish for the transduction reaction. For siRNA-mediated SGK1 knockdown, VM-glia were transfected with siRNA targeting mouse SGK1 (GCTATCTGCACTCCCTAAACA) or control siRNA (scrambled sequence) using Lipofectamine 2000 and 3000 reagents (Thermo Scientific Inc.) according to manufacturer's instructions.

### CM preparation and treatment

Fresh N2 medium or HBSS was added to VM-glial cultures (transduced with sh-Sgk1 or sh-cont), and the GCM was collected every other day for 6 days. The GCMs were adjusted to 0.1–0.15 mg of protein/ml, filtered at 0.45 μm, and stored at − 80°C until use. The GCMs were diluted with N2 medium (1:1, v/v) before adding to the cells in culture.

### RNA-sequencing analysis

RNA-sequencing was carried out at E-biogen (Seoul, Korea). After trimming the reads with a 9 quality score of less than 20 using FastQC and checking the mismatch ratio using Bowtie, all RNA-seq data were mapped to the mouse reference genome (GRCm38/mm10) using STAR. To measure the expression levels of all 46,432 annotated genes, 107,631 transcripts, and 1,276,131 protein-coding (mRNA) records in the mouse genome (based on NCBI RefSeq 13 annotations Release 105: February 2015), we counted reads mapped to the exons of genes using Htseq-count and calculated the fragments per kilobase of exon per million fragments (FPKM) value. Gene classification was based on searches submitted to DAVID (http://david.abcc.ncifcrf.gov/). Gene ontology (GO) and KEGG pathway analyses were performed using DAVID Bioinformatics Resources version 6.8. Heatmap data were generated by MeV 4.9.0. Scatterplot (EXDEGA_v1.6.7, E-biogen) and GSEA (GSEA 4.0.0, Broad Institute, Cambridge, MA, USA) clustering software were used for further analyses.

### Immunostaining

Cultured cells were fixed with 4% paraformaldehyde in PBS, blocked in 0.3–0.6% Triton X-100 with 1% bovine serum albumin for 40–60 minutes, and then incubated with primary antibodies overnight at 4°C. Primary antibody information is shown in Appendix Table S2. Secondary antibodies tagged with Cy3 (1:200, Jackson Immunoresearch Laboratories, West Grove, PA, USA) or Alexa Fluor 488 (1:200, Life Technologies) were used for visualization. The stained cells were mounted using VECTASHIELD with DAPI mounting solution (Vector Laboratories, Burlingame, CA), and images were obtained with an epifluorescence microscope (Leica, Heidelberg, Germany) and confocal microscope (Leica PCS SP5).

To analyze cell maturation, total fiber length, soma size, and intensity, the cells were measured using an image analysis system (Leica LAS).

### Western blotting

Protein was extracted using a RIPA buffer containing a protease inhibitor (Roche, Mannheim, Germany). For cytosolic and nuclear fractionation, cells were prepared using an EpiQuik kit (Epigentek, Brooklyn, NY, USA). Proteins were electrophoresed on SDS–PAGE gel (12% or 4–16% for SNCA aggregate detection), transferred to a nitrocellulose (or polyvinylidene fluoride) membrane, blocked, and then incubated with the primary antibodies listed in Appendix Table S2. Signals were visualized with horseradish peroxidase-conjugated antibodies and captured with ChemiDoc (Bio-Rad, Hercules, CA, USA).

Densitometric quantification of the bands was performed using ImageJ (Image Processing and Analysis in Java, NIH).

### Real-time PCR

Total RNA was prepared using the Trizol reagent (Invitrogen, Carlsbad, CA, USA) through the RNA isolation protocol. cDNA synthesis was carried out using a Superscript kit (Invitrogen). Real-time PCR was performed on a CFX96TM Real-Time System using iQTM SYBR green supermix (Bio-Rad, Hercules, CA, USA), and gene expression levels were determined relative to GAPDH levels.

See Appendix Table S1 for a list of the primers.

### ELISA for IL-1β and IL-6

Cultured cell supernatants were diluted (1:40) and assayed for IL-1β and I-L6 protein levels using enzyme-linked immunosorbent assay (ELISA) kits (R&D Systems, Minneapolis, MN, USA) according to the manufacturer's instructions.

## Mitochondrial assays

Mitochondrial protein dynamics were determined using the integrated portions of young (green) and old (red) MitoTimer proteins (Hernandez et al, 2013). Briefly, cells were transduced with the lentiviruses expressing the pMitoTimer vector (Addgene52659; Addgene, Watertown, MA, USA), and 3–4 days later, the ratio of red to green fluorescence intensities was quantified. Mitochondrial ROS was estimated using MitoSox (Thermo Scientific Inc) and following the manufacturer's protocols. Mitochondrial membrane potential was estimated with a NucleoCounter3000 (NC3000; ChemoMetec, Allerod, Denmark). Intracellular ATP content was measured using an ATP determination kit (Molecular Probes, Eugene, OR, USA).

## Oxidative stress, cell senescence, and viability

Intracellular ROS levels were determined by staining with the ROS indicator 5-(and-6)-chloromethyl-20,70-dichlorodihydrofluorescein diacetate [CM-H2DCF-DA, DCF) (Life Technologies)]. Cell senescence was assessed using an SA-β-gal staining kit (Abcam, Cambridge, UK). In certain cases, the senolytic drugs fisetin and azithromycin were used. Cell viability was assessed with a CCK8 kit (D-Plus™, DonginBio, Seoul, Korea).

## Analyses for SNCA propagation

### Purification of recombinant SNCA and SNCA PFFs
Recombinant SNCA and SNCA fibrils were prepared as described previously (Choi et al, 2015). Briefly, 5 mg/ml monomeric SNCA was incubated at 37°C with continuous agitation at 250 rpm for 2 weeks, sonicated on ice for 3–5 s, and stored at − 80°C until use as SNCA fibrils. The status of SNCA fibrils was determined using the thioflavin T binding assay and an electron microscopic observation per preparation.

### SNCA aggregation analysis
The levels of SNCA aggregate formation were detected in differentiated VM-NPC cultures overexpressing SNCA or treated with SNCA PFFs (1 µg/ml of medium). For SNCA overexpression, VM-NPCs were transduced with lentiviruses expressing human SNCA (pEF1α-α-syn) and differentiated for 15 days. The SNCA oligomers and fibrils were detected with immunocytochemical (p129-SNCA, BioLegend, San Diego, CA, USA) and Western blot (SNCA, BD Biosciences, San Jose, CA, USA) analyses on 20–7% gradient SDS–PAGE, as described above. The levels of SNCA aggregation were also estimated using immunocytochemical analysis for the Bi-FC assay. The VM-NSCs were co-transduced with lentiviruses expressing Venus1-α-syn (V1S; N-terminal of SNCA) and α-syn-Venus2 (SV2; C-terminal of SNCA) and differentiated into mDA neurons for 15 days. Bi-FC-SNCA aggregates were quantified by counting GFP$^+$ puncta.

## Glutamate uptake

Cells were washed twice in tissue buffer (5 mM Tris, 320 mM sucrose, pH 7.4) and exposed to 10 µM glutamate in either Na$^+$-containing Krebs buffer (120 mM NaCl, 25 mM NaHCO$_3$, 5 mM KCl, 2 mM CaCl$_2$, 1 mM KH$_2$PO$_4$, 1 mM MgSO$_4$, and 10% glucose) or Na$^+$-free Krebs (120 mM choline-Cl and 25 mM Tris–HCl) for 10 min at 37°C. Uptake was stopped by placing the cells on ice and washing them twice with wash buffer (5 mM Tris/160 mM NaCl, pH 7.4). Cells were collected and homogenized in 100 µl of assay buffer, and the amount of glutamate in the cell homogenates was measured using a glutamate assay kit (Abcam, Cambridge, MA, USA). Na$^+$-dependent uptake was determined by subtracting Na$^+$-free counts from the total counts in the presence of Na$^+$.

## Determination of intracellular GSH levels

Intra-neuronal GSH levels were determined as previously described (Jeong et al, 2019). Cells were incubated for 2 h with FreSHtracer™ (2 µM, Cell2in, Seoul, Korea), which binds to GSH and emanates fluorescence at 510 nm (peak fluorescence of free form at 580 nm). After washing with D-PBS, fluorescence readings were taken at 580 nm and 510 nm using an Arial II flow cytometer (BD Biosciences, Franklin Lakes, NJ, USA). The GSH levels are expressed as the ratio of the fluorescence at 510 and 580 nm.

## Assay of NLRP3 inflammasome

To induce conventional NLRP3 inflammasome activation, cells were primed with LPS (0.25 − 1 µg/ml, 3 h), followed by ATP treatment (2–2.5 mM, 30–45 min). Inflammasome activation was determined by the presence of active caspase-1 (p10) and active IL-1β in immunoblots from the culture supernatant and by extracellular IL-1β quantification using ELISA (Lee et al, 2019).

## Animal housing

All procedures for animal experiments were approved by the Institutional Animal Care and Use Committee at Hanyang College of Medicine under approval number 2018-0217A. Mice were housed in a specific pathogen-free barrier facility with a 12-h light/dark cycle and maintained on standard chow (5053 PicoLabR Rodent Diet 20). Experiments were performed in accordance with NIH guidelines.

## Preparation of PD mouse models

To generate PD mice by MPTP injection, female ICR mice (10--14 weeks old) received i.p. injections of MPTP (30 mg/kg) once daily for five consecutive days (Jackson-Lewis & Przedborski, 2007). SNCA-induced PD mice were generated by combined treatment with AAV2 expressing human SNCA (α-syn-AAV2) and SNCA PFFs (Thakur et al, 2017). To this end, female mice were anesthetized by Zoletil50 (0.1 mg/kg) mixed with Rompun (93.28-l g/kg), and SNCA PFF (2 µl, 5 mg/ml in PBS) and α-syn-AAV2 (2 µl, $2.1 \times 10^{12}$ GC/ml) were injected bilaterally into the SN (3.3 mm posterior to bregma; ± 1.2 mm lateral to midline;-4.6 mm ventral to dura) (the same volume of PBS injected for control). The infusion was performed at a rate of 0.25 µl per min. The needle (26 gauge) was left in the injection site for 25–30 min after completion of each injection and then removed slowly.

### In vivo SGK1 inhibition in PD model mice

The SGK1 inhibitor GSK-650394 was administered (3 mg/kg, i.p., once a day) beginning a week after SNCA or the last MPTP

injection. In other cases, AAV9 expressing sh-Sgk1 (sh-Sgk1-AAV9) or Sh-control-AAV9 (control) was stereotaxically injected into the SNs of the PD mice one week after the combined SNCA treatment.

## AAV production

AAV9 and 2 expressing sh-SGK1 or SNCA, respectively, under the control of the CMV promoter were generated by subcloning the respective cDNAs into the pAAV-MCS vector (Addgene). Packaging and production of the AAVs (serotypes 2 and 9) was completed by KIST Virus Core (Seoul, Korea). AAV titers were determined with a QuickTiter$^{TM}$ AAV Quantitation Kit (Cell Biolabs).

## Behavior tests

### Pole test
Animals were placed head upwards on top of a vertical wooden pole 50 cm in length (diameter, 1 cm). The base of the pole was placed in the home cage. Once placed on the pole, animals oriented themselves downward and descended the length of the pole back into their home cage. All of the animals received 2 days of training that consisted of five trials for each session. On the test day, animals received five trials, and the time to orient downward was measured.

### Challenging beam traversal test
Motor performance was measured with a novel beam test adapted from traditional beam-walking tests. Briefly, the beam (length, 1 m) started at a width of 3.5 cm and gradually narrowed to 0.5 cm in 1-cm increments. Animals were trained to traverse the length of the beam, starting at the widest section and ending at the narrowest section, for 2 days before real testing. The time required for each animal to traverse the beam was measured.

### Locomotion test
The locomotor activities of the PD mice were monitored using a camera (HD C310, Logitech, Switzerland). Briefly, mice were placed in the center of a 40-cm square cage and allowed to freely explore the apparatus for 20 min while being tracked by a video-recording system. After each test, each mouse was returned to its home cage, and the open field was cleaned with 70% ethyl alcohol and permitted to dry. To assess the process of habituation to the novelty of the arena, mice were exposed to the apparatus for 20 min on two consecutive days for statistical analysis. Total distance and average speed were measured automatically by the software (Stoelting Co., IL, USA).

## Histological analysis

Mice were anesthetized as described above. Brains were sliced at a thickness of 1 mm on a mouse brain slice matrix (ZIVIC Instruments), and 4–6 regions of the SN (ca 1 × 2 mm) were dissected and subjected to real-time PCR analysis. For immunohistochemistry, mice were perfused intracardially with 4% paraformaldehyde in PBS. Brains were removed and immersed in 30% sucrose in PBS until they sank. Midbrain and striatal regions were sliced into 30 μm thicknesses on a freezing microtome (CM 1850; Leica,

## The paper explained

### Problem
Neuronal cells have been the central cell type targeted for drug development in CNS disorders. However, effective disease-modifying therapies that rescue degenerating neurons have not yet been developed. It is mainly because astrocytes and microglia, brain-resident glia, establish harmful inflammatory environments in disease contexts and thereby contribute to the progression of neuronal loss in neurodegenerative disorders. Correcting the diseased properties of glia is therefore an appealing strategy for treating brain diseases.

### Results
In this study, we show that inhibiting glial SGK1 corrects the pro-inflammatory properties of glia by suppressing the intracellular NFκB-, NLRP3-inflammasome, and CGAS-STING-mediated inflammatory pathways. Furthermore, SGK1 inhibition potentiated glial activity to scavenge glutamate toxicity and prevented glial cell senescence and mitochondrial damage. Along with those anti-inflammatory/neurotrophic functions, silencing and pharmacological inhibition of SGK1 protected midbrain dopamine neurons from degeneration and cured pathologic synuclein alpha (SNCA) aggregation and Parkinson disease (PD)-associated behavioral deficits in multiple in vitro and in vivo PD models.

### Impact
Our results provide evidence that SGK1 inhibition in astrocytes and microglia can be therapeutically used in PD. Because the glial pathologies shown to be improved by SGK1 inhibition in this study commonly underlie other CNS disorders, SGK1 inhibition might eventually become a therapeutic intervention that is broadly applicable to a variety of neurodegenerative disorders and neurologic injuries.

Wetzlar, Germany) and subjected to immunofluorescence staining. TH$^+$ mDA neurons in the SN were counted every 8 sections throughout the midbrain, and total SN mDA neuron numbers were calculated. TH$^+$ fiber lengths, the soma sizes of mDA neurons and microglia in the SN, and TH + fiber intensities in the striatum were measured using an image analysis system (Leica LAS).

## Determination of the level of the SGK1 inhibitor GSK-650394 in the brains of PD mice

The level of the SGK1 inhibitor GSK-650394 in the brains of PD mice was determined according to a previous report (Ham et al, 2019). Briefly, the brain tissues were collected 30 min after the last administration of GSK-650394 at a dose of 3 mg/kg. Residual blood contamination was removed by intracardial perfusion with saline. The collected mouse brains were quickly washed with ice cold saline three times and further homogenized in a two-fold volume of PBS on ice. Then, 50 μl of brain homogenate or plasma was deproteinized by adding 100 μl of methanolic solution containing the internal standard (IS) ondansetron. After centrifugation at 14,000 rpm for 15 min at 4°C, the supernatant was injected into the LC-MS/MS system, an Agilent HPLC system (1290 Infinity, Agilent Technologies, Santa Clara, CA) and Agilent 6490 QQQ mass spectrometer with a positive electrospray ionization (ESI$^+$) Agilent Jet Stream ion source (Agilent Technologies, Santa Clara, CA). A good separation of the SGK1 inhibitor and IS from endogenous

substances was achieved using a Synergi 4 μm polar-RP 80A column (150 mm × 2.0 mm, 4 μm, Phenomenex, Torrance, CA). The mobile phase was a mixture of 0.1% formic acid and acetonitrile (20:80) at a flow rate of 0.2 ml/min. The injection volume was 2 μl. For multiple reaction monitoring in the $ESI^+$ mode, the transition of m/z 383.1 → 314.9 and m/z 294.1 → 170 was applied for the SGK1 inhibitor and IS, respectively. The linearity was good, with a weighing of 1/x over the calibration range (5–10,000 ng/ml, R > 0.9944). The observed concentration in the brain is expressed as ng/g of brain.

### Statistical analysis

Animal behavioral assays in this study have been assessed by an independent experimenter blinded to treatment condition. In culture wells (coverslips) and brain sections, immunostained and DAPI-stained cells were counted in 5–20 random areas using an eyepiece grid at a magnification of 200× or 400×, and total positive cells (or percentages) in a well were calculated. All data are expressed as the mean ± SEM, and statistical tests are justified as appropriate. Statistical comparisons were made using the Student's *t*-test (unpaired) or one-way analysis of variance (ANOVA) followed by Bonferroni *post hoc* analysis using SPSS (Statistics 21; IBM Inc. Bentonville, AR). The *n*, *P*-values, and statistical analysis methods are indicated in the figure legends.

## Data availability

The datasets produced in this study are available in the following databases: RNA-seq data: Gene Expression Omnibus: GSE106216 (https://www.ncbi.nlm.nih.gov/geo/query/acc.cgi?acc = GSE106216).

Microarray data: NCBI GEO: GSE145489 (https://www.ncbi.nlm.nih.gov/geo/query/acc.cgi?acc = GSE145489).

**Expanded View** for this article is available online.

### Acknowledgments

This work was supported by the grants 2017R1A5A2015395, 2017M3A9B4062415, and 2020M3A9D8039925, funded by the National Research Foundation of Korea (NRF) of the Ministry of Science and ICT, Republic of Korea. E.-H. J. was supported by the 2020 sabbatical year research grant of the University of Seoul.

### Author contributions

O-CK, J-JS, YY and M-YC designed and performed most experiments; S-HK, JYK and SYK helped with animal experiments; M-JS, IH, JKa, JKi, and M-YC helped with molecular & signaling studies; J-WY, H-JM, E-hJ, HS and M-YC helped with manuscript discussion; O-CK, J-JS, YY and M-YC performed RNA-seq analyses; SL designed and wrote the manuscript.

### Conflict of interest

The authors declare that they have no conflict of interest.

### For more information

i   Information on SGK1: OMIM: https://omim.org/entry/162958.
ii  Information on Parkinson's Disease: https://www.parkinson.org/understanding-parkinsons/what-is-parkinsons.

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
