## [Review Process File · EMBO Molecular Medicine]

Sgk1 inhibition in glia ameliorates pathologies and symptoms of Parkinson's disease

Oh-Chan Kwon, Jae-Jin Song, Yunseon Yang, Seong-Hoon Kim, Ji Young Kim, Min-Jong Seok, Inhwa Hwang, Je-Wook Yu, Jenisha Karmacharya, Han-Joo Maeng, Jiyoung Kim, Eek-hoon Jho, Seung Yeon Ko, Hyeon Son, Mi-Yoon Chang, Sang-Hun Lee

DOI: [10.15252/emmm.202013076](https://doi.org/10.15252/emmm.202013076)

Corresponding authors: Sang-Hun Lee (leesh@hanyang.ac.kr) , Mi-Yoon Chang (mychang@hanyang.ac.kr)

Review Timeline:

Submission Date:	8th Jul 20
Editorial Decision:	30th Jul 20
Revision Received:	3rd Dec 20
Editorial Decision:	24th Dec 20
Revision Received:	20th Jan 21
Accepted:	23rd Jan 21

Editor: Zeljko Durdevic

Transaction Report:

30th Jul 2020

Dear Prof. Lee,

Thank you for the submission of your manuscript to EMBO Molecular Medicine. We have now received feedback from the three reviewers who agreed to evaluate your manuscript. As you will see from the reports below, the referees acknowledge the interest of the study but also raise serious concerns that should be addressed in a major revision. Particular attention should be given to identifying the cell type specific function of SGK1 inhibition. In our opinion, the mechanism of Nurr1/Foxa2-dependent regulation of Sgk1, although relevant, is not a focus of this study and should be addressed only in writing.

Addressing the reviewers' concerns in full will be necessary for further considering the manuscript in our journal, and acceptance of the manuscript will entail a second round of review. EMBO Molecular Medicine encourages a single round of revision only and therefore, acceptance or rejection of the manuscript will depend on the completeness of your responses included in the next, final version of the manuscript. For this reason, and to save you from any frustrations in the end, I would strongly advise against returning an incomplete revision.

We realize that the current situation is exceptional on the account of the COVID-19/SARS-CoV-2 pandemic. Therefore, please let us know if you need more than three months to revise the manuscript.

I look forward to receiving your revised manuscript.

Yours sincerely,

Zeljko Durdevic

***** Reviewer's comments *****

Referee #1 (Remarks for Author):

The manuscript submitted by Kwon et al investigated the effects of Sgk1 inhibition on glia with pro-inflammatory properties and the therapeutic potentials of Sgk1 inhibition using different in vivo models of Parkinson's disease. Their results showed that inhibition of Sgk1 suppressed Kfkb-, NLRP3 inflammasome- and cGAS-STING-mediated inflammatory pathways in the culture system of mixed astrocyte and microglia. In addition, Sgk1 inhibition attenuated glial cell senescence and

mitochondria damage by enhancing glutamate uptake. Lastly, the author found Sgk1 inhibition ameliorated PD-associated behavior and pathology in two PD mouse models.

Overall the authors presented substantial data on the roles of Sgk1 in multiple major cellular pathways in vitro, and the beneficial effect on the inhibition of Sgk1 in PD pathology. However, the data seem a bit too diffuse to form a cohesive story and certain critical questions are not addressed.

Major Comments

1. Extending their previous work (Oh et al. 2016), in this manuscript the authors propose that Nurr1 and Foxa2 are the major regulators of SGK. SGK1 transcription is controlled by a wide variety of inputs including cell stress, hormones, cytosolic Ca²⁺ activity, Insulin, growth factors, and a number of major pathways such as PI3K, mTORC2, and PDK1. However, the mechanisms regarding Nurr1 and Foxa2's regulation of SGK1 - direct or indirect - are not addressed.
2. It remains unclear which major mechanism of SGK1 in vitro contributes to effects on PD pathology in vivo.
3. In vivo experiments (Figures 7-9) using pharmacological and viral manipulation of SGK1 inhibitors do not inform cell-type-specific function.
4. The culture system in this manuscript is a mixture of astrocytes and microglia. Is the ratio of the two cells comparable to that in the brain? Also, the author did not distinguish the roles of Sgk1 in astrocytes and microglia in such an experimental system.
5. According to Figure 8A, the infection efficiency of shSgk1 AAV9 is comparable in neurons and astrocytes: 1. Is it possible the beneficial effects of shSgk1 come from neurons? 2. What are the effects of shSgk1 in astrocytes in the PD models.
6. What is the expression pattern of Sgk1 in both normal brain and PD models? Is it enriched in certain CNS cell types?

Minor Comments

1. In the in vivo interventions (Figures 7-9) the authors do not show SGK1 downregulation after pharmacological and viral interventions. These would be useful controls.
2. For shSgk1 treatment, the efficiency of Sgk1 knockdown should be shown with different cell markers (Astrocyte, microglia, and neuron)
3. There are a number of western blots missing quantification throughout the manuscript.
4. Some figure legends need to be clarified.(e.g. Fig9 Fa dn G: what are the three bars of control or treatment group?)

Referee #2 (Remarks for Author):

This work by Kwon et al reports the underlying mechanism and therapeutic outcome of inhibiting

glial Sgk1 in Parkinson disease models. On the mechanistic front, the authors show an impressive amount of in vitro and in vivo data demonstrating that glial Sgk1 is a key target inhibited by Nurr1/Foxa2 and that inhibition of Sgk1 (by pharmacological or siRNA strategies) suppress proinflammatory pathways and neurotoxic pathways. On the functional/therapeutic front, the authors show that Sgk1 inhibition increases mDA neuronal survival in multiple in vitro models and improves neuronal survival, nigrostriatal projection, and motor performance in two independent PD mouse models. The experiments are well designed and nicely performed. Findings are logically and clearly presented. Overall, this is a very thorough and solid study with a focus of high biological and translational interest. Some revisions and clarifications are recommended.

Overall

- 1) Recommend using HGNO gene symbols for abbreviations, examples, SNCA for alpha-synuclein, TBK1 for Tank binding kinase 1 (P8, L12).
- 2) Current trend uses the non-possessive form for Parkinson disease and Alzheimer disease.
- 3) Unify the symbol for the control treatment group, such as "Cont", throughout all the figure panels. Currently a mixture of terms/symbols were used, creating confusion and ambiguity (e.g., Cont, Cont(-), DMSO, -, _, etc).
- 4) Provide some text clarification on the specificity of Sgk1 inhibitors used in this study based on published information.
- 5) It would be nice if the authors can discuss the potential shared or distinct effects of microglial and astrocytic Sgk1 on neurons.

Specific

- 1) P2, L18: Change to "... that share the common pathology of glia-mediated neuroinflammation."
- 2) Fig. 1C: Explain RC.
- 3) Fig. 1E: Place individual "total protein" panels right under their respective "phosphorylated protein" panels.
- 4) Fig. 1O: Revise the diagram to better illustrate the action of phosphorylated IKKbeta in phosphorylating Ikb and releasing NF-kB for nuclear translocation. The old diagram shows no connection of the nuclear translocated NF-kB to the one in the NF-kB/Ikb complex.
- 5) Fig. 2F: Did the authors draw the graph? If not, I recommend they re-draw it for i) copy right issue, and ii) leave out the molecules not directly relevant to this study.
- 6) Fig. 4F: What is the topographical relationship between the insets relative to the whole pictures? Same, different, or a small part?
- 7) P8, L12: TBK1 instead of phosphor-TANK
- 8) P11, L7: Delete "AD".
- 9) Fig. 5A: Typo (Fig. 6 B&C should be Fig. 5 B&C).
- 10) P12, L10, and Fig. 5D: Explain and provide references for cell-to-cell transfer of SNCA.
- 11) Use TH (red)+DAPI (blue or white) overlaid images for Fig. 6B, 6C, and 6F.
- 12) P13, L4: Typo "pathological aggregation".
- 13) P13, L16: Change "neurodegeneration" to "oxidative and inflammatory damage"
- 14) P13, L25: Typo "neuroprotective/pro-inflammatory"
- 15) P14, L29: Behavioral deficits improved beginning 4 weeks (instead of 3 weeks) after the start of Sgk1 inhibitor treatment.
- 16) Define the time window when locomotor activity tests were performed in Fig. 7D, 7O, and 8G.
- 17) Fig. 9F, 9G: Draw precise lines to indicate which animals showed statistical differences. It is not clear to me whether the horizontal lines with changing positions from panel to panel indicate all 3 or just 1-2 of the animals.
- 18) Fig. 9 Legend (P42, L29 and 30): Typos - it should be (G) instead of (J)

19) Fig. 9 legends (P42, L31): Define how many qPCR repeats for each bar, representing one animal each.

Referee #3 (Remarks for Author):

Neuroinflammation is tightly associated with PD pathogenesis and disease progress. The manuscript entitled "Sgk1 inhibition in glia ameliorates pathologies and symptoms of Parkinson's disease" by Kwon et al. show that Sgk1 has a role in inflammation in both microglia and astrocytes. Inhibition of Sgk1 mediates the anti-inflammatory effects of Nurr1 and Foxa2. Inhibition of Sgk1 also protects DA neurons from α -synuclein(PFF)-induced damage and improves animal behaviors. The study described new findings for understanding the role of Sgk1 in inflammation in association with PD. It is an interesting study. However, there are some weaknesses that should be addressed.

Major points:

1. Fig.1, 1D and 1E showed that Nurr1 or Foxa2 alone decreased Sgk1 expressions. In protein level, the decrease of Sgk1 by Nurr1 or Foxa2 alone is similar to that by the combination of Nurr1 and Foxa2, but the phosphorylations of I κ B and p65 are different among these groups. Also, fig 1E needs to be quantified.
2. Fig. 4, the treatment with hydrogen peroxide is not a way to produce superoxide that is detected by mitoSox. Even in the data sheet of reagent mitoSox, the hydrogen peroxide does not induce mitochondrial superoxide.
3. Fig. 5B, it lacks controls. Treatment of co-cultures with GSK650394 could not exclude the effects of inhibitor on neurons.
4. Fig. 6 and 8, the efficiency of Sgk1 knockdown is not presented by immunoblots or real-time PCR.
5. Fig. 8 A, the amount of microglia is surprisingly high in non-treated brain.
6. Fig. 8, the shRNA with CMV promoter to Sgk1 is not neuronal specific, it still can not rule out the effects on neurons.

Minor point:

1. The title is not correctly described as the study used animal model but not PD patients. It should clearly address in PD animal models.
2. There are many typo errors that should be corrected.

Authors' responses to the editor's comments*The editor's comment:*

Particular attention should be given to identifying the cell type specific function of SGK1 inhibition.

Authors' response: In the revised paper, we added the data regarding the cell type specific expressions of Sgk1 and functions of Sgk1 inhibition as follow:

- 1) Sgk1 expressions in neurons, astrocytes, and microglia in cultures (Suppl. Fig. S6A) and *in vivo* midbrain (Suppl. Fig. S8C).
- 2) Cell type specific functions of Sgk1 inhibition in neurons (Suppl. Fig. S6B), astrocytes, and microglia (Suppl. Fig. S7).
- 3) Cell type specific Sgk1 knockdown data for the mouse midbrain substantia nigra (SN) treated with AAV9-sh-Sgk1 (Fig. 8D and E).

The editor's comment:

In our opinion, the mechanism of Nurr1/Foxa2-dependent regulation of Sgk1, although relevant, is not a focus of this study and should be addressed only in writing.

Authors' response: We very much appreciate this appropriate point and considerate suggestion. Based on this suggestion, we speculated on several possible mechanisms for Nurr1+Foxa2-mediated regulation of Sgk1 expression in the Discussion section (p.20, line 4 - p.20, line 22).

Authors' responses to the reviewers' comments

Referee #1 :

*Portions revised in the text based on the reviewer's points are highlighted in red.

General Summary:

The manuscript submitted by Kwon et al investigated the effects of Sgk1 inhibition on glia with pro-inflammatory properties and the therapeutic potentials of Sgk1 inhibition using different in vivo models of Parkinson's disease. Their results showed that inhibition of Sgk1 suppressed Kfkb-, NLRP3 inflammasome- and cGAS-STING-mediated inflammatory pathways in the culture system of mixed astrocyte and microglia. In addition, Sgk1 inhibition attenuated glial cell senescence and mitochondria damage by enhancing glutamate uptake. Lastly, the author found Sgk1 inhibition ameliorated PD-associated behavior and pathology in two PD mouse models. Overall the authors presented substantial data on the roles of Sgk1 in multiple major cellular pathways in vitro, and the beneficial effect on the inhibition of Sgk1 in PD pathology. However, the data seem a bit too diffuse to form a cohesive story and certain critical questions are not addressed.

Authors' response: We are very grateful for the reviewer's positive comments and suggestions to help enhance the impact of our study. In response to the points and suggestions raised by the reviewer, we have substantially revised the paper. Our point-by-point responses to the reviewer's comments are as follows:

We understand the reviewer's criticism that our study seems a bit too diffuse to form a cohesive story, because we addressed multiple pathogenic manifestations cured by glial Sgk1 inhibition such as inflammation, oxidative and mitochondrial stress, glial cell senescence, α -synucleinopathy, and mDA neuron degeneration. However, as explained in the Discussion section of this manuscript (p.20, line 24 – p.21, line 21 in the revised manuscript), we would say that those pathologic phenomena are closely linked and thus amelioration of the group of inflammatory sequelae was triggered by a single treatment of Sgk1 inhibition in glia. To help this study be more comprehensive, the links between the pathogenic pathways regulated by Sgk1 inhibition are summarized in a schematic representation (Supplementary Fig. S9 in the revised manuscript).

Major Comments

1. Extending their previous work (Oh et al. 2016), in this manuscript the authors propose that Nurr1 and Foxa2 are the major regulators of SGK. SGK1 transcription is controlled by a wide variety of inputs including cell stress, hormones, cytosolic Ca²⁺ activity, Insulin, growth factors, and a number of major pathways such as PI3K, mTORC2, and PDK1. However, the mechanisms regarding Nurr1 and Foxa2's regulation of SGK1 - direct or indirect - are not addressed.

Authors' response: How Nurr1+Foxa2 regulates Sgk1 expression is intriguing but was not addressed in this study.

Nurr1 and Foxa2 have naïve roles as transcription factors that activate gene transcription. Thus, Nurr1+Foxa2 downregulates Sgk1 which is likely mediated in an indirect manner by activating transcriptions of molecules or pathways that can downregulate Sgk1 transcription. As indicated by the reviewer, Sgk1 transcription is controlled by a wide variety of stimuli including cell stress, hormones, cytokines, cytosolic Ca²⁺ activity, and a number of intracellular pathways. Specifically, oxidative stress and DNA damage are common stimulators for Sgk1 transcription (Leong, J Biol Chem., 2003; Lang, Physiol Rev., 2006), while Nurr1 and Foxa2 have been shown to ameliorate those cellular stresses by promoting trophic responses, stress responses, mitochondrial gene expressions as well as DNA repair functions (Sousa, 2007, Stem cells; Song, 2009, Cell stress chaperones; Kadkhodaei, 2013, PNAS; Oh, 2015, EMBO MM; Malewicz, 2011, Genes Dev). This implies that Sgk1 expression is indirectly controlled by Nurr1 and Foxa2 actions to relieve cellular stress. In addition, Nurr1 has been shown to exert an epigenetic repressor function by forming a repressor complex that contains the corepressor element 1 silencing transcription factor complex (CoREST)(Saijo, 2009, Cell). Thus, a Nurr1 (Foxa2)-mediated action that directly inhibits Sgk1 gene transcription cannot be excluded. These speculations are made in the Discussion section of the revised manuscript (p.20, line 6; p.20, line 22).

Due to the complexity in the possible Sgk1 regulatory mechanisms by Nurr1+Foxa2 as described above, we think that the postulated mechanism could not easily be tested within the limited time frame of this paper's revision. In addition, we are afraid to say that the mechanism of Nurr1/Foxa2-dependent regulation of Sgk1, although relevant, is not a focus of this study, and will be addressed in a separate study.

2. It remains unclear which major mechanism of SGK1 in vitro contributes to effects on PD pathology in vivo.

Authors' response: Because all the glial pathogenic pathways regulated by Sgk1 inhibition are closely linked, it is hard to identify which is the major contributing mechanism. We believe that summation of the pathologic pathways mediated by glial Sgk1 inhibition, but not by a single pathology, is responsible for the amelioration of PD pathologies. However, based on our findings, we think that downregulation of the NFκB-mediated inflammatory pathway is positioned at the most upstream point that allows it to initiate and trigger the other therapeutic effects of Sgk1 inhibition. This explanation is provided in the Discussion section of the revised text (p.20, line 24; p.21, line 21 and schematized in Supplementary Fig. S9).

3. In vivo experiments (Figures 7-9) using pharmacological and viral manipulation of SGK1 inhibitors do not inform cell-type-specific function.

Authors' response: As suggested by this reviewer's points 4 and 5, we have carried out cell-type-specific effects experiments to investigate Sgk1 inhibition in pure neuron, astrocyte, and microglia cultures in the revised paper. Expectedly, Sgk1 inhibition in either the astrocytic or microglial cultures resulted in neurotrophic effects that prevented neuronal (mDA neuronal) degeneration (the astrocytic effect was more robust than that of the microglia effect) (Supplementary Fig. S7 and p.15. line 22 – p.16. line 8.). Furthermore, sh-Sgk1-treated astrocytes and microglia cooperated to exert more potent neuroprotective effects. By contrast, Sgk1 inhibition in neuronal cultures non-significantly altered neuronal (mDA neuronal)

viability (**Supplementary. Fig. S6B and p.15, line 5 – 20 in the revised manuscript**). Based on the *in vitro* findings, it is believed that pharmacological inhibition and silencing of Sgk1 in astrocytes and microglia additively/synergistically exerted therapeutic functions in the PD animal models.

4. The culture system in this manuscripts is a mixture of astrocytes and microglia. Is the ratio of the two cells comparable to that in the brain? Also, the author did not distinguish the roles of Sgk1 in astrocytes and microglia in such an experimental system.

Authors' response: The ratio of astrocytes and microglia in the brain varies very much depending on reports, detection methods, and brain regions. Thus, we counted the glial cells and the ratio in the midbrain of adult mice (microglia:19.7±1.7 % of total astrocyte+microglia). The data is now shown in **p. 6, line 10**.)

We appreciate this appropriate point. As described in our response to point 3, cell-type-specific effects of Sgk1 inhibition were carried out in pure astrocyte and microglia cultures. The results suggest astrocyte- and microglia-specific effects of Sgk1 inhibition that act to protect (mDA) neurons from toxic insult. These results have been described in the revised manuscript (**Supplementary Fig. S7 and p.15. line 22 – p.16. line 8**).

5. According to Figure 8A, the infection efficiency of shSgk1 AAV9 is comparable in neurons and astrocytes: 1. Is it possible the beneficial effects of shSgk1 come from neurons? 2. What is the effects of shSgk1 in astrocytes in the PD models.

Authors' response: Our response to this point is same as that in points 3 and 4.

6. What is the expression pattern of Sgk1 in both normal brain and PD models? Is it enriched in certain CNS cell types?

Authors' response: Based on the reviewer point, the following data are shown in the revised text.

- 1> Sgk1 mRNA expression levels in pure neuronal, astrocytic, and microglial cultures (**Supplementary. Fig. S6A**)
- 2> Total and cell-type specific Sgk1 protein expressions in the midbrains of the control and α -syn-PD mice using immunohistochemical analysis for Sgk1+/MAP2 neurons, Sgk1+/GFAP+ astrocytes, and Sgk1+/Iba1+ microglia (**Supplementary. Fig. S8**).

Based on these findings, we concluded that Sgk1 expression increases in the SN of PD mice and the increased Sgk1 expression was ubiquitously detected without a cell-type specificity (**p.17. lines 13–16**).

Minor Comments

1. In the *in vivo* interventions (Figures 7-9) the authors do not show SGK1 downregulation after pharmacological and viral interventions. These would be useful controls.

Authors' response: We appreciate this appropriate point. The data demonstrating total and cell-type-specific Sgk1 expressions that were downregulated in sh-Sgk1-treated mouse midbrain are now shown in Fig. 8D and E of the revised manuscript

2. For shSgk1 treatment, the efficiency of Sgk1 knockdown should be shown with different cell markers (Astrocyte, microglia, and neuron)

Authors' response: Our response to this point is the same as that for minor comment 1.

3. There are a number of western blots missing quantification throughout the manuscript.

Authors' response: Based on the reviewer's comment, all the western blot data were quantified throughout the manuscript.

4. Some figure legends need to be clarified.(e.g. Fig9 Fa dn G: what are the three bars of control or treatment group?)

Authors' response: We are very sorry for the missing information in Fig. 9F and G and the incorrect description in the figure legend. The errors have been corrected in the Figure and its legend (p.46, lines 11–12).

In addition we have carefully check all the figures and their legends.

Referee #2 :

**Portions revised in the text based on the reviewer's points are highlighted in red.*

General Summary: *This work by Kwon et al reports the underlying mechanism and therapeutic outcome of inhibiting glial Sgk1 in Parkinson disease models. On the mechanistic front, the authors show an impressive amount of in vitro and in vivo data demonstrating that glial Sgk1 is a key target inhibited by Nurr1/Foxa2 and that inhibition of Sgk1 (by pharmacological or siRNA strategies) suppress proinflammatory pathways and neurotoxic pathways. On the functional/therapeutic front, the authors show that Sgk1 inhibition increases mDA neuronal survival in multiple in vitro models and improves neuronal survival, nigrostriatal projection, and motor performance in two independent PD mouse models. The experiments are well designed and nicely performed. Findings are logically and clearly presented. Overall, this is a very thorough and solid study with a focus of high biological and translational interest. Some revisions and clarifications are recommended.*

Authors' response: We are very grateful for the reviewer's positive comments and suggestions to help enhance the impact of our study. In response to the points and suggestions raised by the reviewer, we have substantially revised the paper. Our point-by-point responses to the reviewer's comments are as follows:

Overall

1) Recommend using HGNC gene symbols for abbreviations, examples, SNCA for alpha-synuclein, TBK1 for Tank binding kinase 1 (P8, L12).

Authors' response: As recommended, gene symbols were changed to HGNC format throughout the text.

2) Current trend uses the non-possessive form for Parkinson disease and Alzheimer disease.

Authors' response: We appreciate instruction on how to use current descriptive forms. The disease terms have been changed to the non-possessive form throughout the text.

3) Unify the symbol for the control treatment group, such as "Cont", throughout all the figure panels. Currently a mixture of terms/symbols were used, creating confusion and ambiguity (e.g., Cont, Cont(-), DMSO, -, _, etc).

Authors' response: We appreciate the reviewer for carefully checking our manuscript and suggesting correction of the inconsistent terms and symbols in the figures. We have corrected the inconsistencies throughout the figures.

4) Provide some text clarification on the specificity of Sgk1 inhibitors used in this study based on published information.

Authors' response: The specificity for the Sgk1 inhibitors is shown on p.23, line 30; p.23, line 31 of the revised text.

5) It would be nice if the authors can discuss the potential shared or distinct effects of microglial and astrocytic Sgk1 on neurons.

Authors' response: We appreciate this appropriate suggestion. A similar comment was made by the other reviewer. Thus, we have carried out cell-type-specific effects experiments that investigated Sgk1 inhibition in pure astrocyte and microglia cultures in the revised paper. Expectedly, Sgk1 inhibition in either astrocytic or microglial cultures resulted in neurotrophic effects that prevented neuronal (mDA neuronal) degeneration (the astrocytic effect was more robust than the microglia effect). Furthermore, sh-Sgk1-treated astrocytes and microglia cooperated with each other to exert more potent neuroprotective effects. The data are described in Supplementary Fig. S7 and p.16, line 1–8 of the revised manuscript.

Specific

1) P2, L18: Change to "... that share the common pathology of glia-mediated neuroinflammation."

Authors' response: Thank you for suggesting the sentence correction. The sentence was corrected as suggested (p.2, lines 19–20).

2) Fig. 1C: Explain RC.

Authors' response: We are sorry for the missing information. The full name of FC (fold change) and RC (read count) were inserted in Fig. 1B and C and the legend of the revised manuscript (p.40, line 9).

3) Fig. 1E: Place individual "total protein" panels right under their respective "phosphorylated protein" panels.

Authors' response: We rearranged the WB panels as suggested.

4) Fig. 1O: Revise the diagram to better illustrate the action of phosphorylated IKKbeta in phosphorylating Ikb and releasing NF-kB for nuclear translocation. The old diagram shows no connection of the nuclear translocated NF-kB to the one in the NF-kB/Ikb complex.

Authors' response: We revised the schematic summary as suggested in the revised paper.

5) Fig. 2F: Did the authors draw the graph? If not, I recommend they re-draw it for i) copy right issue, and ii) leave out the molecules not directly relevant to this study.

Authors' response: We guess that this comment was regarding Fig. 3F, but not Fig. 2F. We revised the schematic summary in Fig. 3F as suggested.

6) Fig. 4F: What is the topographical relationship between the insets relative to the whole pictures? Same, different, or a small part?

Authors' response: We are sorry for the unclear description. The inset images were taken from the same microscopic fields as the whole pictures. The inset information is now described in the revised text (p.43, lines 14–15 of the Fig. 4F legend).

7) P8, L12: TBK1 instead of phosphor-TANK

Authors' response: We thank the reviewer for carefully checking our manuscript and suggesting we correct this. The typo error has been corrected (p.8, line 18 of the revised manuscript).

8) P11, L7: Delete "AD".

Authors' response: This was deleted (p.11, line 18 of the revised manuscript).

9) Fig. 5A: Typo (Fig. 6 B&C should be Fig. 5 B&C).

Authors' response: The typo error was corrected.

10) P12, L10, and Fig. 5D: Explain and provide references for cell-to-cell transfer of SNCA.

Authors' response: Cell-to-cell transfer of SNCA was described as the major underlying mechanism for the spread of α -synucleinopathy with a representative reference (Luk et al., Science 2012) (p.12, lines 20–21 of the revised manuscript).

The principle of the BiFC system is described on p.12, lines 21–26 of the revised manuscript with a reference (Bae et al., Nature Com., 2014) and a schematic summary (Fig. 5D, lower images).

11) Use TH (red)+DAPI (blue or white) overlaid images for Fig. 6B, 6C, and 6F.

Authors' response: The figures were changed into TH+DAPI overlaid images (Fig. 6B, C, F of the revised manuscript).

12) P13, L4: Typo "pathological aggregation".

Authors' response: This typo was corrected as suggested (p.13, line 20 of the revised manuscript).

13) P13, L16: Change "neurodegeneration" to "oxidative and inflammatory damage"

Authors' response: This was corrected as suggested (p.14, lines 2–3 of the revised manuscript).

14) P13, L25: Typo "neuroprotective/pro-inflammatory"

Authors' response: This was corrected as suggested (p.14, line 12–13 of the revised manuscript).

15) P14, L29: Behavioral deficits improved beginning 4 weeks (instead of 3 weeks) after the start of Sgk1 inhibitor treatment.

Authors' response: We appreciate the reviewer for pointing out this mistake. This was corrected to '4 weeks' (p.16, line 21 of the revised manuscript).

16) Define the time window when locomotor activity tests were performed in Fig. 7D, 7O, and 8G.

Authors' response: The missing information was inserted in the legends of Fig. 7D, 7O and 8G in the revised text (p.45, lines 1–2 and 10; p.45, lines 27–28 of the revised text).

17) Fig. 9F, 9G: Draw precise lines to indicate which animals showed statistical differences. It is not clear to me whether the horizontal lines with changing positions from panel to panel indicate all 3 or just 1-2 of the animals.

Authors' response: This reviewer's point made us realize that the horizontal lines were a bit distorted and inadequately positioned above the bars. In figure 9F and G, statistical comparison of the mRNA values was made between three GSK-650934-treated (red bars) vs three control mice (blue bars); thus, the horizontal lines should have spanned all three bars for each group. This mistake was corrected in revised Fig. 9F, G and the statistical information is described in the legend (p.46, lines 11–12).

18) Fig. 9 Legend (P42, L29 and 30): Typos - it should be (G) instead of (J)

Authors' response: The typo error was corrected (p.46, lines 9–10 of the revised text).

19) Fig. 9 legends (P42, L31): Define how many qPCR repeats for each bar, representing one animal each.

Authors' response: Each bar represents the mean \pm SEM of 3 PCR values from each animal, n=3 animals per group. This information was added to the legend of Fig. 9 in the revised text (p.46, lines 11–12 of the revised text).

Referee #3 (Remarks for Author):

**Portions revised in the text based on the reviewer's points are highlighted in red.*

General Summary: *Neuroinflammation is tightly associated with PD pathogenesis and disease progress. The manuscript entitled "Sgk1 inhibition in glia ameliorates pathologies and symptoms of Parkinson's disease" by Kwon et al. show that Sgk1 has a role in inflammation in both microglia and astrocytes. Inhibition of Sgk1 mediates the anti-inflammatory effects of Nurr1 and Foxa2. Inhibition of Sgk1 also protects DA neurons from α -synuclein(PFF)-induced damage and improves animal behaviors. The study described new findings for understanding the role of Sgk1 in inflammation in association with PD. It is an interesting study. However, there are some weaknesses that should be addressed.*

Authors' response: We are very grateful for the reviewer's positive comments on the value of this study and the valuable suggestions. In response to the points raised by the reviewer, we have substantially revised the paper. Our point-by-point responses to the reviewer's comments are as follows:

Major points:

1. Fig.1, 1D and 1E showed that Nurr1 or Foxa2 alone decreased Sgk1 expressions. In protein level, the decrease of Sgk1 by Nurr1 or Foxa2 alone is similar to that by the combination of Nurr1 and Foxa2, but the phosphorylations of I κ B and p65 are different among these groups. Also, fig 1E needs to be quantified.

Authors' response: The Sgk1 WB band was changed to that exhibiting the greater decrease in Sgk1 protein level by the combined Nurr1+Foxa2 treatment than those by a single treatment with Nurr1 or Foxa2.

2. Fig. 4, the treatment with hydrogen peroxide is not a way to produce superoxide that is detected by mitoSox. Even in the data sheet of reagent mitoSox, the hydrogen peroxide does not induce mitochondrial superoxide.

Authors' response: The author's comment is right. Hydrogen peroxide (H₂O₂) is not directly detected by the mitochondrial superoxide fluorescence indicator (MitoSox). However, we are afraid to say that the purpose of the combined H₂O₂+CCCP (mitochondrial toxin) treatment in this study was to induce mitochondrial damage in glial cultures, and then to assess the effect of Sgk1 inhibitor treatment to relieve (or prevent) the glial cells from incurring mitochondrial damage. With this purpose, glial cell cultures were exposed to a high dose of H₂O₂ (500 μ M)+CCCP (2 μ M) for a relatively long period (3 hr). Along with JC-1 and Mitotimer assays, MitoSox was used to assess the Sgk1 inhibition effect on mitochondrial damage relief. We believe that the treatment sufficiently induced mitochondrial damage (based on previous studies with the data exemplified in Fig.4 in the paper by Wang et al. (Mol Med Rep, 2018) and Fig. 6 in the paper by Weng et al. (Int J Mol Med, 2018), and indeed MitoSox red fluorescence was increased by the toxin treatment in this study (Fig. 4E).

3. Fig. 5B, it lacks controls. Treatment of co-cultures with GSK650394 could not exclude the effects of inhibitor on neurons.

Authors' response: The WB data from a negative control (without exogenous α -synuclein virus+ PFF treatment) is now shown in Supplementary Fig. S3 of the revised text, which shows an efficient induction of α -synuclein aggregation in cultured neuronal cells by the exogenous α -synuclein treatment used in this study (described in p.12, lines 11–14 of the revised manuscript).

The reviewer's comment that suggests the possibility of an Sgk1 inhibitor effect on neuronal cells is very appropriate. Based on this reviewer's point, we examined the Sgk1 inhibitor regulation of α -synucleinopathy in pure neuronal cultures. As result, the levels of monomer and aggregated forms of α -synuclein, and p129- α -syn were indistinguishable in the neuronal cultures treated with the Sgk1 inhibitor compared to those of the inhibitor-untreated controls (Supplementary Fig. S4 of the revised paper). Based on these findings, we could conclude that the Sgk1 inhibitor effect of reducing α -synuclein proteins in the neuron+glia co-cultures (Fig. 5A-C) was mediated by the inhibitor's action on the glia, but not on the neuronal cells (described in p.12, lines 17–20 of the revised manuscript).

4. Fig. 6 and 8, the efficiency of Sgk1 knockdown is not presented by immunoblots or real-time PCR.

Authors' response: We appreciate the reviewer for pointing out the missing data. The Sgk1 knockdown effect in the mouse midbrain injected with sh-Sgk1-AAV9, compared with the sh-control-AAV9, is now shown in Fig. 8D of the revised manuscript. In addition, neuron, astrocyte, and microglia cell type-specific Sgk1 knockdown effects were also estimated in the midbrain sections co-stained with Sgk1/MAP2, Sgk1/GFAP, and Sgk1/Iba-1 (Fig. 8E). The description appears on p.18, lines 14–18 of the revised text.

5. Fig. 8 A, the amount of microglia is surprisingly high in non-treated brain.

Authors' response: The Iba-1/GFP image was replaced by that with lower Iba-1+ microglia abundance.

6. Fig. 8, the shRNA with CMV promoter to Sgk1 is not neuronal specific, it still cannot rule out the effects on neurons.

Authors' response: Indeed, we had the same question raised by another reviewer. Thus we tested the direct effect of Sgk1 inhibition on neuron cell survival very carefully in primary mDA neuronal cultures in which the glial population had been eliminated by Ara-C treatment. Our numerous independent experiments revealed no significant difference in neuronal viability after treating the pure neuronal cultures with the Sgk1 inhibitor. This data was not shown in our original manuscript, but is now described in p.15, lines 11–20 of the revised manuscript and Supplementary Fig. S6.

Minor point:

1. The title is not correctly described as the study used animal model but not PD patients. It should clearly address in PD animal models.

Authors' response: The title has been corrected as suggested (p.1, lines 3–4).

2. There are many typo errors that should be corrected.

Authors' response: Thanks for pointing out typo errors in our manuscript. Three of the authors have carefully checked the text, and corrected typo errors throughout the text.

24th Dec 2020

Dear Prof. Lee,

Thank you for the submission of your revised manuscript to EMBO Molecular Medicine. I am pleased to inform you that we will be able to accept your manuscript pending the following final amendments:

1) With approaching holidays and the end of the year we encountered high number of submissions, so that our data editors were not able to process all received manuscripts before the holiday season. Therefore, we will send you the document with data editor's suggestions after the holidays and as soon as our data editors process your manuscript. Please do not submit your revised manuscript before we send you the file with data editor's suggestions. Thank you for your understanding.

2) In the main manuscript file, please do the following:

- Add up to 5 keywords.
- Remove text colour.
- Make sure that all special characters display well.
- Add callouts for Fig 5F and Fig 7T. There is a callout for Fig 7V but the panel does not exist, please correct.
- In M&M, the statistical paragraph should reflect all information that you have filled in the Authors Checklist, especially regarding randomization, blinding, replication.
- Specify in author contribution initials for Seong-Hoon Kim and Seung Yeon Ko as well as for Jenisha Karmacharya and Jiyoung Kim.
- Indicate in legends $p=$ values, not a range, along with the statistical test used. To keep the figures "clear" some authors found providing an Appendix table Sx with all exact p-values preferable. You are welcome to do this if you want to.
- Correct the reference list. In the reference list, where there are more than 10 authors on a paper, 10 will be listed, followed by "et al.". Please check "Author Guidelines" for more information.
<https://www.embopress.org/page/journal/17574684/authorguide#referencesformat>
- Add data availability statement. Please be aware that all datasets should be made freely available upon acceptance, without restriction. Use the following format to report the accession number of your data:

***** Reviewer's comments *****

Referee #1 (Remarks for Author):

According to our criticisms, the authors improved the manuscript. The findings are novel and the experiments are carefully done.

Referee #2 (Remarks for Author):

The authors have appropriately addressed all the questions and concerns I raised in my previous review.

Referee #3 (Comments on Novelty/Model System for Author):

The revised manuscript addressed my concerns. It was improved and is

Referee #3 (Remarks for Author):

The manuscript was improved. I have no further concern.

The authors performed the requested editorial changes.

23rd Jan 2021

Dear Prof. Lee,

We are pleased to inform you that your manuscript is accepted for publication.

Corresponding Author Name: Sang-Hun Lee

Journal Submitted to: EMBO MOLECULAR MEDICINE

Manuscript Number: EMM-2020-13076-V2